# MEGS²: Memory-Efficient Gaussian Splatting via Spherical Gaussians and Unified Pruning

**Jiarui Chen**[1*]    **Yikeng Chen**[1,2*]    **Yingshuang Zou**[1]    **Ye Huang**[3]    **Peng Wang**[4]
**Yuan Liu**[1†]    **Yujing Sun**[5†]    **Wenping Wang**[6]
[1]HKUST    [2]SZU    [3]SYSU    [4]Adobe    [5]NTU    [6]TAMU

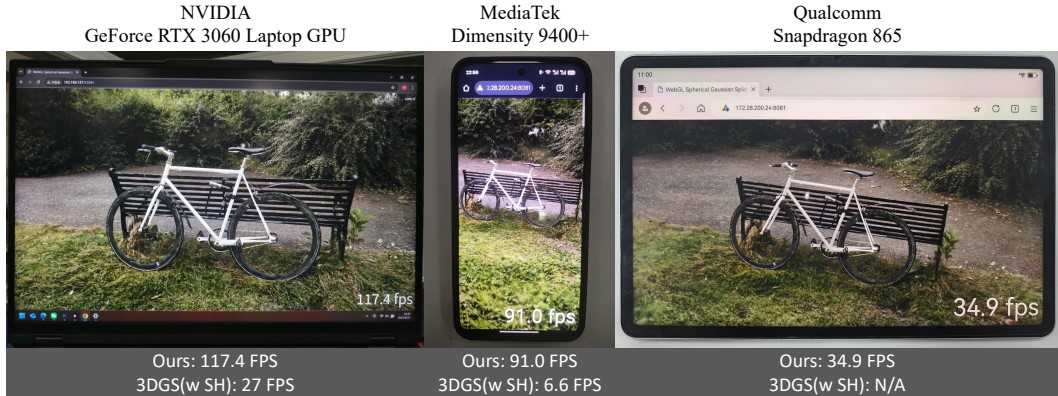

Figure 1: We present **MEGS²**, a memory-efficient framework designed to solve the rendering memory bottleneck of 3D Gaussian Splatting and enable high-quality, real-time rendering on edge devices. As demonstrated in our WebGL-based viewer, 3D Gaussian Splatting(3DGS) with Spherical Harmonics (SH) exhibits low frame rates on desktop GPU and fails to run on some mobile platforms. In contrast, MEGS² achieves interactive frame rates across all tested devices, significantly expanding the applicability of 3DGS. The detailed result is in Appendix A.3 and A.4.

## Abstract

3D Gaussian Splatting (3DGS) has emerged as a dominant novel-view synthesis technique, but its high memory consumption severely limits its applicability on edge devices. A growing number of 3DGS compression methods have been proposed to make 3DGS more efficient, yet most only focus on storage compression and fail to address the critical bottleneck of rendering memory. To address this problem, we introduce **MEGS²**, a novel memory-efficient framework that tackles this challenge by jointly optimizing two key factors: the total primitive number and the parameters per primitive, achieving unprecedented memory compression. Specifically, we fully replace the memory-intensive Spherical Harmonics with lightweight, arbitrarily oriented and prunable Spherical Gaussian lobes as our color representations. More importantly, we propose a unified soft pruning framework that models primitive-number and lobe-number pruning as a single constrained optimization problem. Experiments show that MEGS² achieves a 50% static VRAM reduction and a **40%** rendering VRAM reduction compared to existing methods, while maintaining comparable rendering quality.

## 1 Introduction

Although 3D Gaussian Splatting (3DGS) (Kerbl et al., 2023) has been rapidly replacing implicit neural radiance fields (NeRF) (Mildenhall et al., 2020) as the dominant paradigm in neural rendering, thanks to its fast reconstruction, real-time performance, and high-quality outputs, supporting its application across devices with varying constraints has become increasingly important. As a result,

---

*Equal contribution.
†Corresponding authors

compression of 3DGS (Bagdasarian et al., 2025) is gaining significantly more attention, with the goal of enabling real-world use cases on edge devices, such as mobile 3D scanning and previewing, virtual try-on, and real-time rendering in video games.

Nevertheless, existing compression methods only focus on storage compression rather than memory compression. While the former storage compression speeds up the one-time data transfer of 3DGS files, the rendering memory dictates whether the 3DGS rendering process can run smoothly on edge devices. This limits the applicability of 3DGS applications to low-end devices like cell phones. For example, some methods based on neural compression, vector quantization, and hash grid compression (Chen et al., 2024; Girish et al., 2024; Lee et al., 2024) achieve high storage compression rates. However, these methods require decoding the full Gaussian parameters from a compressed state before rendering, even resulting in a larger rendering memory than the 3DGS methods without compression. Another branch of works, i.e., primitive pruning methods (Zhang et al., 2025b; Fang & Wang, 2024), is effective at reducing both storage and rendering memory due to the reduced primitives. However, these pruning methods still need to retain a substantial number of 3D Gaussians for a reasonable rendering quality, which still consumes a large rendering memory. How to further reduce the rendering memory of 3DGS-based rendering methods remains an open question.

In this paper, we observe that the overall memory consumption for 3DGS rendering is intrinsically tied to two key factors: the total primitive count and the parameters per primitive. Thus, we propose a novel framework that simultaneously reduces both factors, rather than only optimizing for the primitive numbers in previous methods Zhang et al. (2025b); Fang & Wang (2024). Specifically, the rendering VRAM can be divided into a static component and a dynamic component. The static part relates directly to the total number of Gaussian primitives loaded into the renderer, which is a product of the primitive count and the parameters per primitive. On the other hand, the dynamic part, which consists of intermediate data like projected 2D Gaussian parameters and the tile-depth-gaussian key-value table, is also related to the primitive count in the specific camera viewpoint. This inherent relationship underscores that to reduce the overall memory footprint, we need to reduce both the number of primitives and the per-primitive parameter size.

To address the memory bottleneck, we must reduce both primitive counts and per-primitive parameters. We first analyze the limitations of using Spherical Harmonics (SH) for color representation. While effective for low-frequency lighting, SH functions are inherently global and require many high-order coefficients to represent localized, high-frequency details like sharp highlights. This results in low parameter utilization and makes them difficult to compress due to the varying parameter counts across different degrees. While SG-Splatting (Wang et al., 2024a) pioneered using Spherical Gaussians (SG) to mitigate these issues, they relied on a hybrid model combining SG with SH. Advancing in this direction, we introducefully standalone, arbitrarily-oriented, and prunable SG as a more memory-efficient alternative. Such representation excel at modeling view-dependent signals with very few parameters, and their complexity can be flexibly controlled by the number of lobes. This inherent locality and sparsity make it a convenient and effective choice for compression.

Based on the SG-based representation, we further introduce a novel unified soft pruning framework. We leverage the favorable properties of Spherical Gaussians to dynamically prune redundant lobes for each primitive, thereby compressing the per-primitive parameter count. To achieve a globally optimal memory footprint, our framework models the two traditionally separate pruning problems—primitive-count pruning and per-primitive lobe pruning—as a single constrained optimization problem, with the total parameter budget serving as a unified constraint. Extensive experiments demonstrate that the proposed MEGS$^2$ achieves an excellent balance between rendering quality and VRAM efficiency.

Overall, our contributions can be summarized as follows:

- Advancing beyond the hybrid approach in SG-Splatting (Wang et al., 2024a), we introduce a fully standalone color representation by replacing Spherical Harmonics entirely with arbitrarily-oriented and prunable Spherical Gaussians. This significantly reduces the per-primitive parameter count and thus lowers rendering VRAM with minimal impact on quality.
- We propose a unified soft pruning framework that models both primitive-count and lobe-count as a memory-constrained optimization problem, which yields superior performance compared to existing staged or hard-pruning methods.
- We achieve unprecedented memory compression for 3DGS, surpassing both vanilla and state-of-the-art lightweight methods. Our method delivers over an **8×** static VRAM compression and

nearly a **6**× rendering VRAM compression compared to vanilla 3DGS, while maintaining or even improving rendering quality. Furthermore, it still achieves a **2**× static VRAM compression and a 40% rendering VRAM reduction over the SOTA method, GaussianSpa, with comparable quality.

## 2 RELATED WORK

### 2.1 EVOLUTION OF SPLATTING-BASED SCENE REPRESENTATIONS

While Neural Radiance Fields (NeRF) (Mildenhall et al., 2020) and their variants (Barron et al., 2021; Müller et al., 2022; Barron et al., 2023) achieved excellent quality in novel view synthesis, their slow rendering speeds precluded real-time applications. 3D Gaussian Splatting (3DGS) (Kerbl et al., 2023) overcame this limitation by introducing a differentiable rasterizer for 3D Gaussians, enabling unprecedented real-time performance with high visual fidelity. The success of 3DGS, however, highlighted its primary challenge: a substantial memory and storage footprint. This has directly motivated the body of work on 3DGS compression and pruning that we discuss subsequently.

### 2.2 MEMORY ANALYSIS IN 3DGS RENDERING

In most 3DGS compression studies (Bagdasarian et al., 2025), the rendering memory footprint has not been thoroughly analyzed or compared. Memory usage can be conceptualized into two main components: a static portion, consisting of the total parameters of all loaded Gaussian primitives, and a dynamic portion, representing the runtime overhead. The dynamic overhead, which is highly dependent on a renderer's implementation, includes the storage of preprocessed 2D Gaussian attributes—an overhead that scales significantly with the number of rendering channels, as seen in multi-channel applications like Feature Splatting (Qiu et al., 2024). Additionally, it includes data structures required by tile-based renderers, an overhead first discussed and optimized by FlashGS (Feng et al., 2024) for large-scale scenes.

### 2.3 PRUNING TECHNIQUES FOR 3DGS

Pruning unimportant primitives is a natural and widely explored approach for 3DGS compression. Some work has focused on refining adaptive density control strategies to achieve more efficient representations (Kheradmand et al., 2025; Cheng et al., 2024; Liu et al., 2024a; Pateux et al., 2025; Mallick et al., 2024; Kim et al., 2024). Other research has concentrated on defining better importance metrics to decide which primitives to remove, with notable examples including LP-3DGS (Zhang et al., 2024), mini-splatting (Fang & Wang, 2024), and Reduced3DGS (Papantonakis et al., 2024), the latter of which also involves spherical harmonic pruning. A recent noteworthy development is GaussianSpa (Zhang et al., 2025b), which models pruning as a constrained optimization problem, offering a novel perspective and good compatibility with other pruning techniques. While effective at reducing the number of primitives, these methods generally achieve a limited compression ratio and are therefore often used as an initial step within a larger, integrated compression pipeline.

### 2.4 OTHER COMPRESSION SCHEMES FOR 3DGS

Besides pruning, researchers have applied common compression techniques like vector quantization, scalar quantization, neural, and hash grid compression to 3DGS (Girish et al., 2024; Lee et al., 2024; Fan et al., 2024; Lee et al., 2025; Liu et al., 2024b; Navaneet et al., 2024; Xie et al., 2024; Shin et al., 2025). While methods with entropy encoding (Chen et al., 2024; 2025; Liu et al., 2025a; Wang et al., 2024b) achieve the highest storage compression ratios, they are often limited to structured (anchor-based) Gaussian representations (Lu et al., 2023; Ren et al., 2024; Zhang et al., 2025a). These techniques may drastically reduce file size, but they fail to deliver a comparable reduction in rendering memory. This is because they cannot render directly from a compressed state; instead, they must first decode the full Gaussian parameters. The resulting memory footprint is, therefore, never smaller than that required for uncompressed Gaussian primitives, and the decoding process itself, particularly when involving neural networks, can introduce significant additional memory overhead.

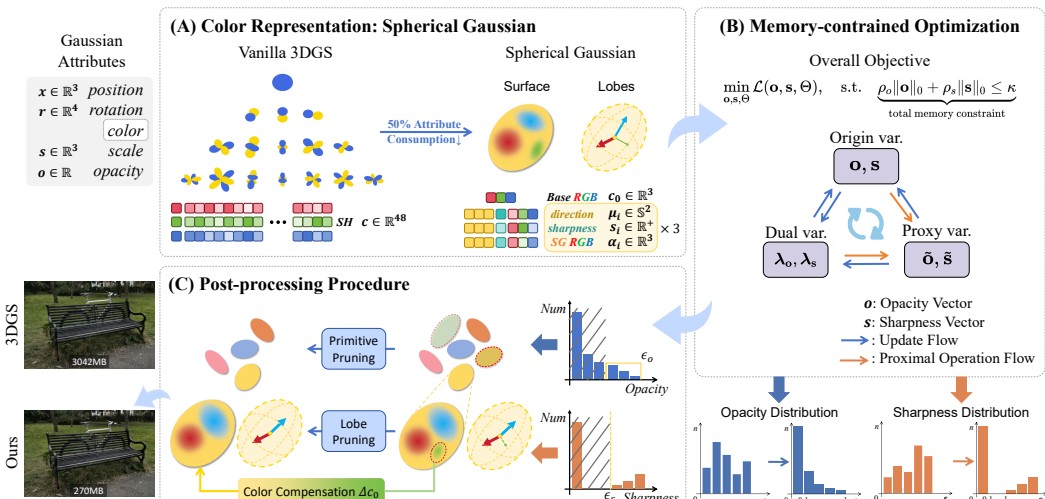

Figure 2: **Overview of our proposed MEGS².** (A) In Section 3.2, we first replace the Spherical Harmonics with arbitrarily-oriented and prunable Spherical Gaussians. (B) In Section 3.3.1, we formulate the compression as a memory-constrained optimization problem, which is solved using an ADMM-inspired approach (Section 3.3.2). (C) In Section 3.3.3, near-invalid primitives and low-sharpness lobes are removed, and a color compensation term (Eq.12) is introduced to recover the energy of the removed lobes.

## 3 METHOD

We present MEGS², **M**emory-**E**fficient **G**aussian **S**platting via **S**pherical **G**aussian and Unified Pruning, a novel Gaussian compression framework designed from a VRAM-centric perspective. Our method strikes a balance between rendering quality and memory footprint by simultaneously optimizing both the number of Gaussian primitives and the average parameters per primitive.

### 3.1 PRELIMINARIES FOR 3D GAUSSIAN SPLATTING

3D Gaussian Splatting (3DGS) (Kerbl et al., 2023) represents a scene using a set of 3D Gaussian primitives. Each primitive is defined by a center $\mu \in \mathbb{R}^3$, a covariance matrix $\Sigma \in \mathbb{R}^{3 \times 3}$, a scalar opacity $o$, and a view-dependent color model. The color is determined by a function $c(\mathbf{v}) : \mathbb{S}^2 \to \mathbb{R}^3$, which maps a viewing direction $\mathbf{v}$ to an RGB color. In the original work, this function is modeled using Spherical Harmonics (SH).

To render an image, these 3D primitives are projected onto the 2D image plane. The final pixel color is computed by alpha-blending the primitives that overlap with the pixel in front-to-back order, according to the equation: $C = \sum_i c_i \alpha_i \prod_{j=1}^{i-1}(1 - \alpha_j)$. Here, the rendering opacity $\alpha_i$ is a product of the primitive's scalar opacity $o_i$ and its 2D Gaussian value at the pixel location.

### 3.2 SPHERICAL GAUSSIANS

Although the original 3D Gaussian Splatting (3DGS) utilized Spherical Harmonics (SH) for color modeling, as we discussed in Section 1, SH functions suffer from limitations in parameter efficiency and the ability to model local and high-frequency signals.

In contrast, Spherical Gaussians (SG) offer a more compact representation, where the parameter count can be controlled by adjusting the number of lobes. As we demonstrated in Figure 4, a three-lobe SG requires only about half the parameters of a 3rd-order SH while achieving comparable expressive power and superior high-frequency detail capture because, in most cases, we only need to model highlights. Furthermore, as shown in Table 2, SG is more amenable to pruning than SH.

SG was first introduced in real-time rendering to support all-frequency shadows from both point lights and environment lights (Wang et al., 2009). Specifically, a lobe of SG has the form

$$G(\mathbf{v}; \mu, s, a) = ae^{s(\mu \cdot \mathbf{v} - 1)}, \tag{1}$$

where $\mu \in \mathbb{S}^2$ is the unit-length lobe axis, $s \in \mathbb{R}^+$ controls the sharpness, and $a \in \mathbb{R}^3$ is the RGB amplitude vector, with $\mathbf{v} \in \mathbb{S}^2$ being the viewing direction.

We use the sum of all lobes of SG for color modeling and introduce a diffuse term to model the direction-independent component. The view-dependent color $c(\mathbf{v})$ is thus computed as:

$$c(\mathbf{v}) = c_0 + \sum_{i=1}^{n} G(\mathbf{v}; \mu_i, s_i, a_i). \tag{2}$$

where $c(\mathbf{v}) \in \mathbb{R}^3$ is the final color from the viewing direction $\mathbf{v} \in \mathbb{S}^2$, $c_0 \in \mathbb{R}^3$ is the direction-independent diffuse color, and the sum is over $n$ all lobes of SG, with $\mu_i \in \mathbb{S}^2$ being the lobe axis, $s_i \in \mathbb{R}^+$ the sharpness, and $a_i \in \mathbb{R}^3$ the RGB amplitude for the $i$-th spherical lobe.

**Choice of arbitrarily-oriented SG lobes**  It is particularly notable that the lobe axes of different SG are not constrained to be orthogonal, nor is any regularization term introduced to enforce orthogonality. Instead, each SG lobe is allowed to have an arbitrary direction. This flexibility grants the SG model a higher degree of freedom, leading to greater representation capability compared to models with fixed orthogonal axes. As shown in Figure 4 (in Appendix), SG-Splatting (Wang et al., 2024a) with fixed orthogonal axes demonstrated significant rendering performance degradation, further underscoring the importance of supporting arbitrarily oriented SG lobes.

### 3.3 UNIFIED SOFT-PRUNING FRAMEWORK

Benefiting from the property of Spherical Gaussians to control the number of parameters with the number of lobes, they provide an ideal object for pruning. Given that most Gaussian primitives in a scene require only a few lobes to effectively model their color, and that the number of Gaussian primitives is itself often redundant, we propose a novel unified soft pruning framework. In the framework, we first redefine the pruning of both Gaussian primitives and spherical lobes as a unified optimization problem with a total memory-overhead constraint (Section 3.3.1). To handle the non-differentiable nature of this constraint, we introduce an ADMM-inspired (Boyd et al., 2011) algorithm to efficiently solve the problem (Section 3.3.2). After optimization, the model enters a post-processing procedure where the final primitives and spherical lobes are removed, with rendering quality subsequently recovered through a color compensation strategy and minor fine-tuning (Section 3.3.3).

### 3.3.1 PROBLEM FORMULATION

We unify primitive count pruning and spherical lobe pruning into a memory-constrained optimization problem. The $L_0$ norm of the opacity vector represents the number of active primitives, as primitives with zero opacity do not contribute to rendering. Similarly, the $L_0$ norm of the sharpness vector denotes the number of active spherical lobes, since lobes with zero sharpness exhibit no view-dependent effect and their color can be directly added to the diffuse term $c_0$. Building upon GaussianSpa (Zhang et al., 2025b)'s sparsification framework and our analysis, we extend the pruning objective from primitive count optimization to total memory budget control, formalized as:

$$\min_{\mathbf{o}, \mathbf{s}, \Theta} \mathcal{L}(\mathbf{o}, \mathbf{s}, \Theta), \quad \text{s.t.} \quad \underbrace{\rho_o \|\mathbf{o}\|_0 + \rho_s \|\mathbf{s}\|_0 \leq \kappa}_{\text{total memory constraint}} \tag{3}$$

where $\mathbf{o} \in \mathbb{R}^{N \times 1}$ means opacity vector for $N$ Gaussian primitives, $\mathbf{s} \in \mathbb{R}^{N \times 3}$ means flattened sharpness vector for $N$ Gaussians, $\Theta \in \mathbb{R}^{N \times 13}$ (other Gaussian variables), $\rho_o = 11$ and $\rho_s = 7$ count base parameters for a single Gaussian primitive and a single SG lobe respectively, and $\kappa$ is the total parameter budget. $\mathcal{L}(\mathbf{o}, \mathbf{s}, \Theta)$ is the reconstruction loss function.

### 3.3.2 MEMORY-CONSTRAINED OPTIMIZATION

In this subsection, we aim to solve the memory-constrained optimization defined in Section 3.3.1. Due to the introduction of a non-differentiable component (the $L_0$ norm) into the constraint, we can-

---

**Algorithm 1** Memory-constrained optimizing

---

**Require:**
  Initial parameters: $\Theta^0, \mathbf{o}^0, \mathbf{s}^0$
  Proxy variables: $\tilde{\mathbf{o}}^0 = \mathbf{o}^0, \tilde{\mathbf{s}}^0 = \mathbf{s}^0$
  Dual variables: $\boldsymbol{\lambda}_{\mathbf{o}}^0 = \mathbf{0}, \boldsymbol{\lambda}_{\mathbf{s}}^0 = \mathbf{0}$
  Learning rate $\eta$, penalty $\delta_o, \delta_s$, budget $\kappa$
**Ensure:**
  Optimized parameters: $\Theta^K, \mathbf{o}^K, \mathbf{s}^K$
 1: **for** $k = 0$ **to** $K - 1$ **do**
 2:  **Gradient Step:**
 3:  $\Theta^{k+1} \leftarrow \Theta^k - \eta \nabla_{\Theta} \mathcal{L}(\Theta^k, \mathbf{o}^k, \mathbf{s}^k)$
 4:  $\mathbf{o}^{k+1} \leftarrow \mathbf{o}^k - \eta \left[ \nabla_{\mathbf{o}} \mathcal{L}(\Theta^k, \mathbf{o}^k, \mathbf{s}^k) + \delta_o(\mathbf{o}^k - \tilde{\mathbf{o}}^k + \boldsymbol{\lambda}_{\mathbf{o}}^k) \right]$
 5:  $\mathbf{s}^{k+1} \leftarrow \mathbf{s}^k - \eta \left[ \nabla_{\mathbf{s}} \mathcal{L}(\Theta^k, \mathbf{o}^{k+1}, \mathbf{s}^k) + \delta_s(\mathbf{s}^k - \tilde{\mathbf{s}}^k + \boldsymbol{\lambda}_{\mathbf{s}}^k) \right]$
 6:  **Proximal Step:**
 7:  $(\tilde{\mathbf{o}}^{k+1}, \tilde{\mathbf{s}}^{k+1}) \leftarrow \mathbf{prox}_h(\mathbf{o}^{k+1} + \boldsymbol{\lambda}_{\mathbf{o}}^k, \mathbf{s}^{k+1} + \boldsymbol{\lambda}_{\mathbf{s}}^k)$
 8:  **Dual Update:**
 9:  $\boldsymbol{\lambda}_{\mathbf{o}}^{k+1} \leftarrow \boldsymbol{\lambda}_{\mathbf{o}}^k + (\mathbf{o}^{k+1} - \tilde{\mathbf{o}}^{k+1})$
10:  $\boldsymbol{\lambda}_{\mathbf{s}}^{k+1} \leftarrow \boldsymbol{\lambda}_{\mathbf{s}}^k + (\mathbf{s}^{k+1} - \tilde{\mathbf{s}}^{k+1})$
11: **end for**
12: **Return** $\Theta^K, \mathbf{o}^K, \mathbf{s}^K$

---

not directly optimize it with regularized stochastic gradient descent. Instead, we adopt an ADMM-inspired approach by introducing proxy variables $\tilde{\mathbf{o}}$ and $\tilde{\mathbf{s}}$ for $\mathbf{o}$ and $\mathbf{s}$, respectively. This decomposition strategy enables us to split the original optimization problem into two tractable subproblems:

$$\min_{\mathbf{o}, \mathbf{s}, \Theta} \mathcal{L}(\mathbf{o}, \mathbf{s}, \Theta) + \frac{\delta}{2} \left( \rho_o \|\mathbf{o} - \tilde{\mathbf{o}} + \boldsymbol{\lambda}_{\mathbf{o}}\|^2 + \rho_s \|\mathbf{s} - \tilde{\mathbf{s}} + \boldsymbol{\lambda}_{\mathbf{s}}\|^2 \right) \tag{4}$$

$$\min_{\tilde{\mathbf{o}}, \tilde{\mathbf{s}}} h(\tilde{\mathbf{o}}, \tilde{\mathbf{s}}) + \frac{\delta}{2} \left( \rho_o \|\mathbf{o} - \tilde{\mathbf{o}} + \boldsymbol{\lambda}_{\mathbf{o}}\|^2 + \rho_s \|\mathbf{s} - \tilde{\mathbf{s}} + \boldsymbol{\lambda}_{\mathbf{s}}\|^2 \right) \tag{5}$$

where $h(\cdot, \cdot)$ is an indicator function enforcing the sparsity constraint:

$$h(\mathbf{o}, \mathbf{s}) = \begin{cases} 0 & \text{if } \rho_o \|\mathbf{o}\|_0 + \rho_s \|\mathbf{s}\|_0 \leq \kappa, \\ +\infty & \text{otherwise} \end{cases} \tag{6}$$

and the proxy variables are updated via the proximal operator:

$$(\tilde{\mathbf{o}}, \tilde{\mathbf{s}}) = \mathbf{prox}_h(\mathbf{o} + \boldsymbol{\lambda}_{\mathbf{o}}, \mathbf{s} + \boldsymbol{\lambda}_{\mathbf{s}}). \tag{7}$$

Here, $\boldsymbol{\lambda}_{\mathbf{o}}$ and $\boldsymbol{\lambda}_{\mathbf{s}}$ denote the corresponding Lagrange multipliers for the constraints. The specific algorithm flow can be found in Algorithm 1, and the detailed derivation is in Appendix A.1.1.

**Proximal Operator Implementation** The proximal operator projects $\mathbf{o}$ and $\mathbf{s}$ onto the constraint $\rho_o \|\mathbf{o}\|_0 + \rho_s \|\mathbf{s}\|_0 \leq \kappa$, compatible with any importance metric. To simplify the design and leverage importance criteria from prior primitive-pruning research, we choose to factorize the projection:

$$\tilde{\mathbf{o}} = \mathbf{prox}_h(\mathbf{o} + \boldsymbol{\lambda}_{\mathbf{o}}), \|\tilde{\mathbf{o}}\|_0 < \kappa_{\mathbf{o}} \tag{8}$$

$$\tilde{\mathbf{s}} = \mathbf{prox}_h(\mathbf{s} + \boldsymbol{\lambda}_{\mathbf{s}}), \|\tilde{\mathbf{s}}\|_0 < \kappa_{\mathbf{s}} \tag{9}$$

For opacity projection, Zhang et al. (2025b) has established multiple proximal operators that are selected based on scene characteristics. Our approach reuses these operators (See Appendix A.2).

For sharpness projection, we project the attribute onto the constraint space by retaining only the $\kappa_s$ most important spherical lobes. This retention is based on a dynamic range metric, which quantifies the view-dependent color change contributed by the $i$-th lobe:

$$D_i = |\max_{\mathbf{v}}(c_i) - \min_{\mathbf{v}}(c_i)| = |a_i|(1 - e^{-2s_i}) \tag{10}$$

where $a_i$ is the RGB amplitude for the $i$-th spherical lobe, and $\kappa_s$ controls the total number of active lobes. The $\kappa_s$ elements corresponding to the highest $D_i$ values in $(\mathbf{s} + \boldsymbol{\lambda}_{\mathbf{s}})$ are preserved.

**Discussion: Sequential vs. Unified Pruning** While sequential pruning tackles memory reduction in a two-stage process—first reducing primitive count, then per-primitive parameters—our unified framework jointly optimizes both factors as a single problem. This approach finds a better trade-off between primitive count and per-primitive complexity, thus avoiding the sub-optimal solutions often found by sequential methods. For a detailed experimental validation, please refer to Section 4.4.2.

### 3.3.3 Post-Processing Procedure

After completing the constrained optimization process, we obtain a large number of primitives and spherical lobes with near-zero opacity and sharpness values, respectively. As the hard constraints are enforced on the proxy variables rather than the original ones, these near-zero values must still be pruned to achieve an actual memory and storage benefit. We address this with a three-step post-processing strategy: first, we remove Gaussian primitives whose opacity falls below a certain threshold. Then, for spherical lobes, we remove those with negligible sharpness and introduce a simple color compensation method to minimize the average per-view color variation and mitigate performance degradation caused by their removal.

This compensation is achieved by finding an optimal term, $\Delta c_0$, that minimizes the integral of the squared color difference over all view directions on the unit sphere:

$$\min_{\Delta c_0} \int_{\mathbb{S}^2} \left((c_0 + \Delta c_0) - (c_0 + G(\mathbf{v}; \mu_i, s_i, a_i))\right)^2 d\mathbf{v} \tag{11}$$

Solving equation 11 yields an exact compensation term that preserves the color of the removed lobe:

$$\Delta c_0 = a_i \cdot \frac{1 - e^{-2s_i}}{2s_i} \tag{12}$$

The diffuse term of the parent primitive is then updated with this value:

$$c_0' = c_0 + \Delta c_0 \tag{13}$$

Finally, after removing some primitives and spherical lobes, we continue to fine-tune for a small number of steps to recover the rendering quality. The detailed derivation is in Appendix A.1.2. Figure 5 presents the distribution of the spherical lobes after post-processing procedure across scenes.

## 4 Experiments

### 4.1 Experimental Settings

**Datasets and Metrics**  Following 3DGS (Kerbl et al., 2023), we evaluate the rendering performance on real-world datasets, including all indoor scenes and outdoor scenes in Mip-NeRF360 (Barron et al., 2022), two scenes from Tanks & Temples (Knapitsch et al., 2017), and two scenes from Deep Blending (Hedman et al., 2018). Additionally, we measure the VRAM overhead during the Gaussian loading (static) and rendering stages on these scenes.

**Baselines**  On all datasets, we benchmark our method against four categories of baselines: (1) vanilla 3DGS (Kerbl et al., 2023) and SG-Splatting (Wang et al., 2024a); (2) lightweight 3DGS methods based on primitive pruning: LP-3DGS (Zhang et al., 2024), Mini-Splatting (Fang & Wang, 2024), MaskGaussian (Liu et al., 2025b) and GaussianSpa (Zhang et al., 2025b); (3) lightweight 3DGS methods based on pruning spherical harmonic coefficients: Reduced3DGS (Papantonakis et al., 2024); (4) 3DGS compression methods utilizing orthogonal techniques such as neural compression and vector quantization: CompactGaussian (Lee et al., 2024), EAGLES (Girish et al., 2024), LightGaussian (Fan et al., 2024), LocoGS (Shin et al., 2025) and MesonGS (Xie et al., 2024).

**Experimental Setup**  To ensure a fair comparison, we follow the core training procedures established by 3DGS (Kerbl et al., 2023) and GaussianSpa (Zhang et al., 2025b). We evaluate rendering quality using PSNR, SSIM, and LPIPS, and measure efficiency via static and rendering VRAM consumption. For a detailed description of our specific implementation, as well as the definitions and measurement procedures for both static and rendering VRAM, please refer to Appendix A.2.

### 4.2 Main Results and Analysis

The quantitative results are summarized in Table 1, comparing our approach with several existing lightweight 3DGS methods. Overall, our method achieves comparable or slightly superior rendering quality to the current state-of-the-art lightweight 3DGS methods, while demonstrating significantly lower VRAM overhead across all existing methods. Compared to vanilla 3DGS, our method achieves a 0.4 dB PSNR improvement on the DeepBlending dataset and a 0.012 LPIPS reduction on

Table 1: **Quantitative comparison across three datasets.** Best results are in red region , second best are in orange region . Memory (VRAM) values are in MB. HQ denotes the version prioritizing high rendering quality, and LM denotes the version prioritizing lower VRAM consumption.

| Method | Mip-NeRF 360 | | | | | Tanks&Temples | | | | | DeepBlending | | | | |
|---|---|---|---|---|---|---|---|---|---|---|---|---|---|---|---|
| | PSNR↑ | SSIM↑ | LPIPS↓ | VRAM↓ Stat. | Rend. | PSNR↑ | SSIM↑ | LPIPS↓ | VRAM↓ Stat. | Rend. | PSNR↑ | SSIM↑ | LPIPS↓ | VRAM↓ Stat. | Rend. |
| 3DGS | 27.48 | 0.813 | 0.217 | 648 | 1717 | 23.68 | 0.849 | 0.171 | 370 | 1021 | 29.71 | 0.902 | 0.242 | 582 | 1569 |
| SG-Splatting | 27.27 | 0.813 | 0.218 | 416 | 1327 | 23.47 | 0.840 | 0.177 | 229 | 822 | 29.57 | 0.901 | 0.247 | 357 | 983 |
| Reduced3DGS | 27.21 | 0.810 | 0.225 | 191 | 493 | 23.51 | 0.839 | 0.187 | 90 | 252 | 29.60 | 0.902 | 0.248 | 121 | 340 |
| CompactGaussian | 27.08 | 0.798 | 0.247 | 267 | 838 | 23.32 | 0.831 | 0.201 | 177 | 469 | 29.79 | 0.901 | 0.258 | 187 | 532 |
| EAGLES | 27.23 | 0.810 | 0.240 | – | – | 23.37 | 0.840 | 0.200 | 333 | 965 | 29.86 | 0.910 | 0.250 | 510 | 1550 |
| LightGaussian | 27.13 | 0.806 | 0.237 | 290 | 640 | 23.44 | 0.832 | 0.202 | 168 | 437 | – | – | – | – | – |
| LocoGS | 27.28 | 0.809 | 0.231 | 609 | 994 | 23.62 | 0.846 | 0.182 | 458 | 732 | 30.05 | 0.905 | 0.247 | 643 | 1040 |
| MesonGS-FT | 26.98 | 0.801 | 0.233 | 709 | 1336 | 23.32 | 0.837 | 0.193 | 424 | 783 | 29.51 | 0.901 | 0.251 | 656 | 1202 |
| LP-3DGS | 27.12 | 0.805 | 0.239 | 420 | 1239 | 23.41 | 0.834 | 0.198 | 251 | 769 | – | – | – | – | – |
| Mini-Splatting | 27.40 | 0.821 | 0.219 | 125 | 477 | 23.45 | 0.841 | 0.186 | 72 | 253 | 30.05 | 0.909 | 0.254 | 89 | 324 |
| MaskGaussian | 27.43 | 0.811 | 0.227 | 271 | 799 | 23.72 | 0.847 | 0.181 | 132 | 517 | 29.69 | 0.907 | 0.244 | 156 | 501 |
| GaussianSpa | 27.56 | 0.824 | 0.215 | 115 | 448 | 23.73 | 0.857 | 0.162 | 106 | 336 | 30.00 | 0.912 | 0.239 | 104 | 372 |
| Ours(HQ) | 27.54 | 0.824 | 0.209 | 55 | 265 | 23.45 | 0.853 | 0.159 | 51 | 211 | 30.17 | 0.912 | 0.233 | 54 | 243 |
| Ours(LM) | 27.21 | 0.814 | 0.227 | 40 | 224 | 23.27 | 0.851 | 0.167 | 37 | 163 | 30.01 | 0.908 | 0.246 | 33 | 193 |

the Tanks & Temples dataset. Furthermore, it achieves more than an **8×** compression rate for static VRAM and nearly a **6×** compression rate for rendering VRAM across all datasets. Even when compared to the current state-of-the-art lightweight 3DGS method, GaussianSpa (Zhang et al., 2025b), we still achieve a nearly 2× compression rate for static VRAM on the Mip-NeRF360 dataset and reduce rendering VRAM by approximately 40% with comparable rendering quality.

**Analysis on Existing Compression Methods** Many recent methods achieve high storage compression but fail to reduce VRAM. Techniques based on hash grid compression, vector quantization or neural networks (e.g., CompactGaussian (Lee et al., 2024), EAGLES (Girish et al., 2024)) must first decompress their parameters into a renderable state. This requirement, coupled with the decoding process itself, results in a VRAM footprint that can exceed even vanilla 3DGS (Appendix, Tab. 3). Similarly, HAC++ (Chen et al., 2025), a SOTA method designed specifically for anchor-based 3DGS, is not a general solution and offers minimal VRAM reduction for the same reason. In contrast, our method achieves superior perceptual quality (SSIM/LPIPS) with 50-60% less VRAM and a 1.5-1.7x rendering speedup.

**Analysis on Pruning-based Methods** Pruning-based methods (Zhang et al., 2025b; Fang & Wang, 2024; Zhang et al., 2024) effectively reduce VRAM but face a bottleneck, as aggressively reducing primitive count degrades quality. Our method overcomes this by not only pruning primitives but also reducing per-primitive costs. When compared to methods that also compress color, such as Reduced3DGS (Papantonakis et al., 2024) which prunes SH coefficients, our advantage is twofold: the superior amenability of our Spherical Gaussians to pruning, and our more effective unified soft pruning framework, as validated in Section 4.4.

## 4.3 QUALITATIVE RESULTS

Figure 3 presents a qualitative comparison of MEGS[2] against baselines, including GaussianSpa and 3DGS on various scenes. In Bicycle and Truck, our method accurately recovers specular reflections on smooth surfaces and mirrors, details that other methods fail to fully capture. In Bonsai, MEGS[2] faithfully captures high-contrast lighting in both contours and brightness. Furthermore, MEGS[2] delivers a cleaner and more complete reconstruction in Playroom. Notably, MEGS[2] achieves these high-quality results with a VRAM footprint of only 50-60% compared to GaussianSpa, demonstrating a substantial enhancement in memory efficiency. These visual results indicate that our Spherical Gaussian based representation has a stronger capability for fitting local view-dependent signals than Spherical Harmonics, leading to sharper and more photorealistic images. More qualitative results are in Appendix, Figure 7.

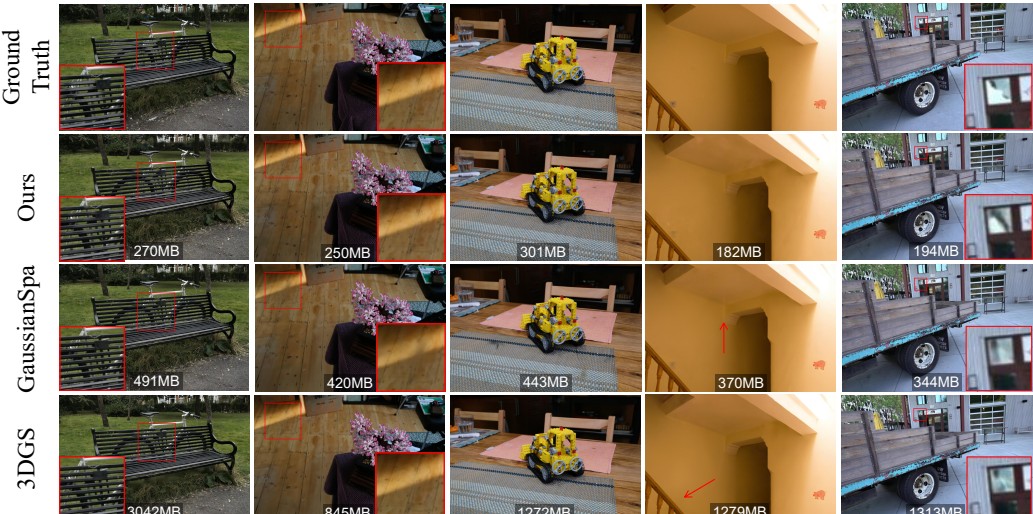

Figure 3: Qualitative results on the Bicycle, Bonsai, Kitchen, Playroom and Truck scenes comparing to previous methods and the corresponding ground truth images from test views. The rendering VRAM consumption for the corresponding method is annotated at the bottom of each image.

### 4.4 ABLATION STUDIES

In this section, we conduct a series of ablation studies to validate the effectiveness of our proposed designs and answer two key questions: 1. Is the replacement of spherical harmonics with spherical Gaussians necessary? And why not use the orthogonal-axis spherical variants (fixed-axis SG)? (Section 4.4.1) 2. Is our proposed unified soft pruning framework effective? And does it offer a significant advantage over simply combining existing pruning methods? (Section 4.4.2)

#### 4.4.1 ABLATION ON COLOR MODELING

To validate our color model, we replace the spherical harmonics (SH) in 3DGS with our arbitrarily-oriented spherical Gaussians (SG) while keeping training settings identical. Our SG model achieves a superior quality-VRAM trade-off (see Figure 4). Furthermore, we show that the fixed-axis SG from SG-Splatting (Wang et al., 2024a) struggles to capture complex view-dependent effects, leading to a sharp quality drop (about 0.6 dB PSNR). This rigidity also makes it incompatible with our lobe-pruning strategy.

#### 4.4.2 ABLATION ON PRUNING STRATEGIES

We evaluate our unified soft pruning framework against several ablations and baselines, with results in Table 2. First, replacing our soft pruning with a hard-pruning variant (soft → hard) that only optimizes opacity results in a significant performance drop. Second, performing primitive and lobe pruning sequentially (unified → sequential) is inferior to our joint optimization, confirming the advantage of a unified approach. We also found that removing our color compensation harms performance (w/o color

Table 2: **Ablation studies for different pruning strategies** on Mip-NeRF360 dataset.

| Method | PSNR↑ | SSIM↑ | LPIPS↓ | VRAM↓ |
|---|---|---|---|---|
| Spa+Redu. | 26.05 | 0.776 | 0.280 | 402 |
| Spa(SH → SG) | 27.01 | 0.808 | 0.230 | 339 |
| soft → hard | 27.23 | 0.814 | 0.228 | 288 |
| unified → seq. | 27.33 | 0.818 | 0.222 | 328 |
| w/o color comp. | 27.46 | 0.822 | 0.213 | **265** |
| full model | **27.54** | **0.824** | **0.209** | **265** |

comp.). Finally, we test two naive baselines. Combining GaussianSpa with Reduced3DGS leads to severe quality degradation, showing that existing SH pruning metrics fail on fewer primitives. Simply replacing SH with SG in GaussianSpa is also outperformed by our method. This demonstrates that our pruning strategy not only reduces memory but also acts as a regularizer, improving rendering quality. Visual comparison are in Appendix, Figure 9.

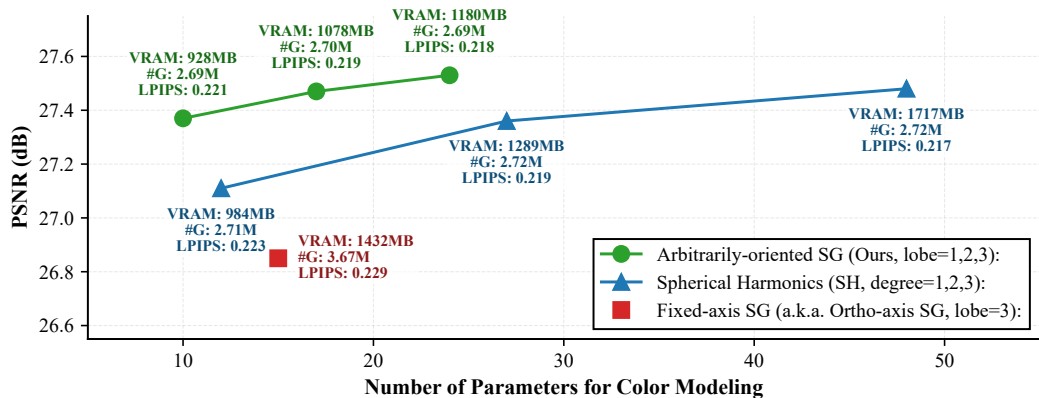

Figure 4: Comparison of different 3DGS color representations on Mip-NeRF360 dataset.

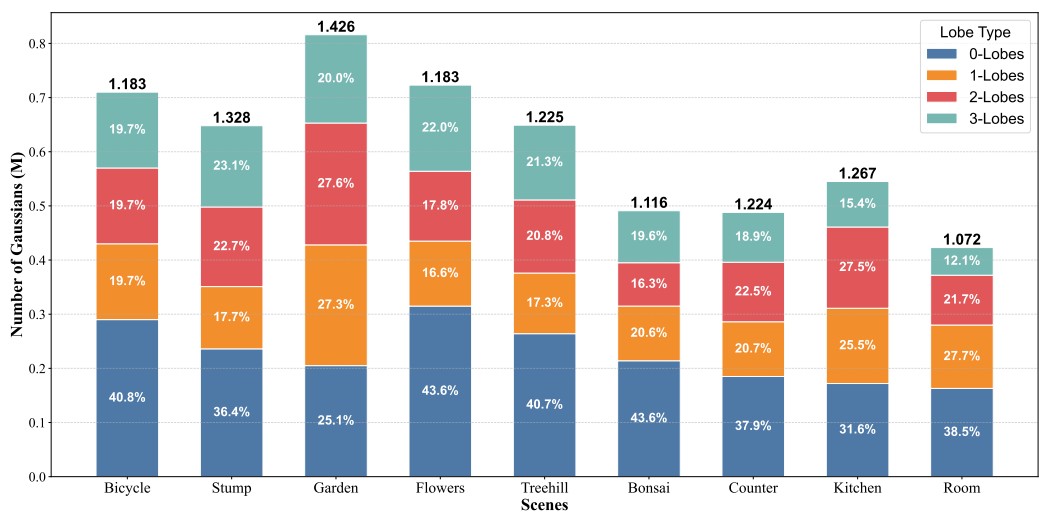

Figure 5: Distribution of Gaussian lobes across scenes.
We report the distribution of Gaussians using 0 to 3 lobes for each scene in the Mip-NeRF 360 dataset. The value above each bar represents the average lobe count per Gaussian.

## 5 CONCLUSION

We introduced MEGS[2], a framework that addresses the 3D Gaussian Splatting memory bottleneck by jointly optimizing primitive count and per-primitive parameters. Our approach combines a lightweight spherical Gaussian color representation with a unified soft pruning method to achieve state-of-the-art memory compression. MEGS² shifts the focus of 3DGS compression towards rendering VRAM efficiency, paving the way for high-quality rendering on edge devices.

**Limitation** Our method focuses on compressing static memory for broad, renderer-agnostic applicability. We leave the optimization of implementation-specific dynamic VRAM for future work, and note that the model's performance on highly complex highlights warrants further investigation.

### ACKNOWLEDGMENTS

This work is supported by the donation from Kerry Group Limited in "30-for-30" Talent Acquisition Campaign of HKUST.

### ETHICS STATEMENT

Our work presents no direct ethical concerns. The primary application of our method is for novel view synthesis from captured data.

REPRODUCIBILITY STATEMENT

To ensure reproducibility: (1) While not included with this submission, our full project (including the WebGL renderer, training, and evaluation scripts) will be released on GitHub upon publication. (2) All experimental details, including our VRAM measurement protocol, are provided in Appendix A.2. (3) Our data and preprocessing follow the official 3DGS implementation. (4) Derivations and proofs for our core algorithms are in Appendix A.1.1 and A.1.2.

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

# A APPENDIX

## A.1 ALGORITHMIC DERIVATION

### A.1.1 DERIVATION FOR MEMORY-CONSTRAINED OPTIMIZING

This appendix provides the detailed derivation of Algorithm 1 presented in the main text. The algorithm solves the memory-constrained optimization problem with $L_0$ norm constraints using an ADMM-inspired approach.

**Problem Formulation**  Consider the constrained optimization problem:

$$
\begin{aligned}
\min_{\mathbf{o},\mathbf{s},\Theta} \quad & \mathcal{L}(\mathbf{o},\mathbf{s},\Theta) \\
\text{s.t.} \quad & \rho_o\|\mathbf{o}\|_0 + \rho_s\|\mathbf{s}\|_0 \leq \kappa
\end{aligned}
\tag{A.1}
$$

where $\mathcal{L}$ is the loss function, $\|\cdot\|_0$ denotes the $L_0$ norm (count of non-zero elements), and $\kappa$ represents the memory budget.

**Augmented Lagrangian Formulation**  Introduce proxy variables $\tilde{\mathbf{o}}, \tilde{\mathbf{s}}$ to reformulate the problem:

$$
\begin{aligned}
\min_{\mathbf{o},\mathbf{s},\Theta,\tilde{\mathbf{o}},\tilde{\mathbf{s}}} \quad & \mathcal{L}(\mathbf{o},\mathbf{s},\Theta) \\
\text{s.t.} \quad & \mathbf{o} = \tilde{\mathbf{o}}, \quad \mathbf{s} = \tilde{\mathbf{s}} \\
& \rho_o\|\tilde{\mathbf{o}}\|_0 + \rho_s\|\tilde{\mathbf{s}}\|_0 \leq \kappa
\end{aligned}
\tag{A.2}
$$

The augmented Lagrangian function is constructed as:

$$
\begin{aligned}
\mathcal{L}_\delta(\mathbf{o},\mathbf{s},\Theta,\tilde{\mathbf{o}},\tilde{\mathbf{s}},\boldsymbol{\lambda_o},\boldsymbol{\lambda_s}) =& \mathcal{L}(\mathbf{o},\mathbf{s},\Theta) \\
& + \boldsymbol{\lambda_o}^\top(\mathbf{o} - \tilde{\mathbf{o}}) + \boldsymbol{\lambda_s}^\top(\mathbf{s} - \tilde{\mathbf{s}}) \\
& + \frac{\delta}{2}\left(\rho_o\|\mathbf{o} - \tilde{\mathbf{o}}\|^2 + \rho_s\|\mathbf{s} - \tilde{\mathbf{s}}\|^2\right)
\end{aligned}
\tag{A.3}
$$

where $\boldsymbol{\lambda_o}$ and $\boldsymbol{\lambda_s}$ are Lagrange multipliers, and $\delta > 0$ is the penalty parameter.

**ADMM Alternating Minimization**  The ADMM framework decomposes the optimization into three subproblems:

**Primal Variables Update**  With proxy variables and multipliers fixed, update the primal variables:

$$
\begin{aligned}
(\Theta^{k+1},\mathbf{o}^{k+1},\mathbf{s}^{k+1}) =& \arg\min_{\Theta,\mathbf{o},\mathbf{s}} \mathcal{L}(\mathbf{o},\mathbf{s},\Theta) \\
& + \frac{\delta}{2}\left[\rho_o\|\mathbf{o} - \tilde{\mathbf{o}}^k + \boldsymbol{\lambda_o}^k\|^2 + \rho_s\|\mathbf{s} - \tilde{\mathbf{s}}^k + \boldsymbol{\lambda_s}^k\|^2\right]
\end{aligned}
\tag{A.4}
$$

Using gradient descent approximation:

$$
\begin{aligned}
\Theta^{k+1} &= \Theta^k - \eta\nabla_\Theta\mathcal{L}(\Theta^k,\mathbf{o}^k,\mathbf{s}^k) \\
\mathbf{o}^{k+1} &= \mathbf{o}^k - \eta\left[\nabla_\mathbf{o}\mathcal{L}(\Theta^k,\mathbf{o}^k,\mathbf{s}^k) + \delta_o(\mathbf{o}^k - \tilde{\mathbf{o}}^k + \boldsymbol{\lambda_o}^k)\right] \\
\mathbf{s}^{k+1} &= \mathbf{s}^k - \eta\left[\nabla_\mathbf{s}\mathcal{L}(\Theta^k,\mathbf{o}^{k+1},\mathbf{s}^k) + \delta_s(\mathbf{s}^k - \tilde{\mathbf{s}}^k + \boldsymbol{\lambda_s}^k)\right]
\end{aligned}
\tag{A.5}
$$

where $\delta_o = \delta\rho_o, \delta_s = \delta\rho_s$.

**Proxy Variables Update**  With primal variables and multipliers fixed, update the proxy variables:

$$
(\tilde{\mathbf{o}}^{k+1},\tilde{\mathbf{s}}^{k+1}) = \arg\min_{\tilde{\mathbf{o}},\tilde{\mathbf{s}}} h(\tilde{\mathbf{o}},\tilde{\mathbf{s}}) + \frac{\delta}{2}\left[\rho_o\|\mathbf{o}^{k+1} - \tilde{\mathbf{o}} + \boldsymbol{\lambda_o}^k\|^2 + \rho_s\|\mathbf{s}^{k+1} - \tilde{\mathbf{s}} + \boldsymbol{\lambda_s}^k\|^2\right]
\tag{A.6}
$$

where $h(\tilde{\mathbf{o}}, \tilde{\mathbf{s}})$ is the indicator function enforcing the sparsity constraint:

$$h(\tilde{\mathbf{o}}, \tilde{\mathbf{s}}) = \begin{cases} 0 & \text{if } \rho_o \|\tilde{\mathbf{o}}\|_0 + \rho_s \|\tilde{\mathbf{s}}\|_0 \leq \kappa \\ +\infty & \text{otherwise} \end{cases} \tag{A.7}$$

The solution is given by the proximal operator:

$$(\tilde{\mathbf{o}}^{k+1}, \tilde{\mathbf{s}}^{k+1}) = \mathbf{prox}_h(\mathbf{o}^{k+1} + \boldsymbol{\lambda}_{\mathbf{o}}^k, \mathbf{s}^{k+1} + \boldsymbol{\lambda}_{\mathbf{s}}^k) \tag{A.8}$$

**Dual Variables Update** Update the Lagrange multipliers:

$$\begin{aligned} \boldsymbol{\lambda}_{\mathbf{o}}^{k+1} &= \boldsymbol{\lambda}_{\mathbf{o}}^k + (\mathbf{o}^{k+1} - \tilde{\mathbf{o}}^{k+1}) \\ \boldsymbol{\lambda}_{\mathbf{s}}^{k+1} &= \boldsymbol{\lambda}_{\mathbf{s}}^k + (\mathbf{s}^{k+1} - \tilde{\mathbf{s}}^{k+1}) \end{aligned} \tag{A.9}$$

### A.1.2 Derivation for Color Compensation

$$\min_{\Delta c_0} \int_{\mathbb{S}^2} \left( (c_0 + \Delta c_0) - (c_0 + G(\mathbf{v}; \boldsymbol{\mu}_i, s_i, a_i)) \right)^2 d\mathbf{v} \tag{A.10}$$

where $G(\mathbf{v}; \boldsymbol{\mu}, s, a) = a e^{s(\boldsymbol{\mu} \cdot \mathbf{v} - 1)}$, $\boldsymbol{\mu} \in \mathbb{S}^2$ is the unit-length lobe axis, $s \in \mathbb{R}^+$ controls the sharpness, and $a \in \mathbb{R}^3$ is the RGB amplitude vector, with $\mathbf{v} \in \mathbb{S}^2$ being the viewing direction.

Simplifying the objective function:

$$\min_{\Delta c_0} \int_{\mathbb{S}^2} \left( \Delta c_0 - G(\mathbf{v}; \boldsymbol{\mu}_i, s_i, a_i) \right)^2 d\mathbf{v} \tag{A.11}$$

Expanding the squared term:

$$\min_{\Delta c_0} \int_{\mathbb{S}^2} \left( \Delta c_0^2 - 2\Delta c_0 G(\mathbf{v}; \boldsymbol{\mu}_i, s_i, a_i) + G(\mathbf{v}; \boldsymbol{\mu}_i, s_i, a_i)^2 \right) d\mathbf{v} \tag{A.12}$$

Taking derivative with respect to $\Delta c_0$ and setting to zero:

$$\frac{\partial}{\partial \Delta c_0} = 2 \int_{\mathbb{S}^2} \left( \Delta c_0 - G(\mathbf{v}; \boldsymbol{\mu}_i, s_i, a_i) \right) d\mathbf{v} = 0 \tag{A.13}$$

Solving for the optimal $\Delta c_0$:

$$\Delta c_0 = \frac{\int_{\mathbb{S}^2} G(\mathbf{v}; \boldsymbol{\mu}i, s_i, a_i) d\mathbf{v}}{\int_{\mathbb{S}^2} d\mathbf{v}} = \frac{1}{4\pi} \int_{\mathbb{S}^2} G(\mathbf{v}; \boldsymbol{\mu}_i, s_i, a_i) d\mathbf{v} \tag{A.14}$$

Evaluating the spherical Gaussian integral. Without loss of generality, align $\boldsymbol{\mu}_i$ with the z-axis:

$$\begin{aligned} \int_{\mathbb{S}^2} G(\mathbf{v}; \boldsymbol{\mu}_i, s_i, a_i) \, d\mathbf{v} &= a_i \int_0^{2\pi} \int_0^{\pi} e^{s_i(\cos\theta - 1)} \sin\theta \, d\theta \, d\phi \\ &= 2\pi a_i \int_0^{\pi} e^{s_i(\cos\theta - 1)} \sin\theta \, d\theta \\ &\quad (\text{substitute } u = \cos\theta, du = -\sin\theta d\theta) \\ &= 2\pi a_i \int_{-1}^{1} e^{s_i(u-1)} \, du \\ &= 2\pi a_i \cdot \frac{1 - e^{-2s_i}}{s_i}. \end{aligned}$$

Substituting back:

$$\Delta c_0 = \frac{1}{4\pi} \cdot 2\pi a_i \cdot \frac{1 - e^{-2s_i}}{s_i} = a_i \cdot \frac{1 - e^{-2s_i}}{2s_i} \tag{A.15}$$

The diffuse term is updated as:

$$c_0' = c_0 + \Delta c_0 \tag{A.16}$$

## A.2 EXPERIMENTAL DETAILS

**Implementation Details**  Our experiments are conducted on a single NVIDIA RTX 3090 GPU (24GB). For a fair comparison, we adopt the same hyperparameters and optimizer used by vanilla 3DGS (Kerbl et al., 2023) on all datasets. In terms of the hyperparameters for our unified soft pruning framework, we conduct the memory-constrained optimization for a total of 10,000 iterations, applied at an interval of 50 iterations. We set the penalty parameter $\delta$ to 0.0005. For the post-processing procedure defined in Section 3.3.3, we set the sharpness threshold to 1 to prune spherical lobes with negligible contribution. Regarding the training strategy, we employ the same densification, pruning, and fine-tuning procedures as GaussianSpa, conducting densification training, sparse training, and fine-tuning for the same number of steps. To ensure the accuracy of our VRAM measurements, we reproduce 3DGS, GaussianSpa (Zhang et al., 2025b), EAGLES (Girish et al., 2024), CompactGaussian (Lee et al., 2024), LightGaussian (Fan et al., 2024), LocoGS (Shin et al., 2025), MesonGS (Xie et al., 2024), and Reduced3DGS (Papantonakis et al., 2024) for direct VRAM measurement. For other pruning-based methods, we utilize both the rendering quality metrics and primitive counts reported in the GaussianSpa paper. We then estimate their corresponding VRAM consumption based on our own VRAM measurements for 3DGS and GaussianSpa. EAGLES cannot be trained on a single NVIDIA RTX 3090 GPU with 24GB on some scenes in Mip-NeRF360, so we do not report VRAM consumption. For fairness, MesonGS uses the version that continues fine-tuning after compression, and LocoGS utilizes point clouds from COLMAP for initialization to maintain consistency with other methods.

**Evaluation Details**  For rendering performance, we report the peak signal-to-noise ratio (PSNR), structural similarity (SSIM), and learned perceptual image patch similarity(LPIPS). For VRAM consumption, we report both static VRAM overhead and rendering VRAM overhead. The former refers to the VRAM required to load all 3DGS primitives into the renderer and dequantize or decode them into a ready-to-render state. This is not necessarily equal to the file size, as some methods might represent Gaussian attributes in a more compact or lower-precision data format but still require restoring them to full, 32-bit precision Gaussian attributes before rendering. The latter denotes the peak VRAM usage during the rendering process, which is typically larger than the static VRAM due to the introduction of intermediate variables (e.g., projected 2D Gaussian attributes, key-value tables for tile-based rendering) during rendering. We measure this value across all test viewpoints and report the average. To exclude the overhead introduced by the framework itself and the impact of memory fragmentation, we note that the memory allocation and management of the official 3DGS implementation's renderer are almost entirely handled by PyTorch. Therefore, all VRAM-related metrics are reported using the values provided by the PyTorch framework.

**Proximal Operators Implementation**  As mentioned in the main text, for opacity projection, we reuse the two proximal operators from GaussianSpa (Zhang et al., 2025b). These operators select $\kappa_o$ primitives to preserve based on different importance criteria. The first is a **Magnitude-Based Selection**, which sorts the input vector $(\mathbf{o} + \boldsymbol{\lambda_o})$ and preserves the $\kappa_o$ elements with the highest magnitudes. The second operator employs an **Importance-Based Selection**, adopting the importance score metric from MiniSplatting (Fang & Wang, 2024). The importance score $I_i$ for the $i$-th Gaussian is the sum of its blending weights across all intersecting camera rays seen during training:

$$I_i = \sum_{j=1}^{K} w_{ij} \tag{A.17}$$

where the blending weight $w_{ij}$ is the product of the Gaussian's opacity $\alpha_i$, its projected 2D value $G_i^{2D}$, and the accumulated transmittance $T_{ij}$ along the $j$-th ray. Here, $K$ is the total number of intersected rays. The operator then preserves the $\kappa_o$ primitives with the highest importance scores. While GaussianSpa selects the specific operator based on scene characteristics (e.g., indoor vs. outdoor), we simplify the overall framework by consistently adopting the **Importance-Based Selection** as our unified approach across all scenes.

## A.3 MORE QUANTITIVE RESULTS

Table 3: **Average parameter costs for the color model per Gaussian primitive.** Storage and Rendering costs are measured in the number of float32 parameters. Decode Overhead indicates whether a method introduces significant additional VRAM during calculating color. The values of our method are calculated on the DeepBlending dataset.

| Method | Storage | Rendering | Decode Overhead |
|---|---|---|---|
| 3DGS | 48 | 48 | No |
| EAGLES | $< 1$ | $> 48$ | Yes |
| CompactGaussian | $< 1$ | $> 32$ | Yes |
| ours w/o lobe-pruning | 24 | 24 | No |
| ours | 9.7 | 9.7 | No |

Table 4: **Comparison of anchor-based 3DGS and specialized SOTA compression schemes** on the Mip-NeRF360 dataset. The best result is shown in **bold**.

| Method | PSNR↑ | SSIM↑ | LPIPS↓ | VRAM↓ | FPS↑ |
|---|---|---|---|---|---|
| Scaffold-GS | 27.74 | 0.811 | 0.226 | 612 | 123 |
| HAC++ (highrate) | **27.81** | 0.811 | 0.231 | 637 | 115 |
| HAC++ (lowrate) | 27.60 | 0.803 | 0.253 | 514 | 132 |
| Ours | 27.54 | **0.824** | **0.209** | **265** | **200** |

Table 5: **Mip-NeRF360 Indoor per scene results.** Best results are in red region , second best are in orange region . Memory (VRAM) values are in MB.

| Scene | Method | PSNR↑ | SSIM↑ | LPIPS↓ | Static VRAM↓ | Rendering VRAM↓ |
|---|---|---|---|---|---|---|
| Bonsai | 3DGS | 32.23 | 0.940 | 0.205 | 393 | 798 |
| | Reduced3DGS | 31.44 | 0.933 | 0.214 | 66 | 276 |
| | GaussianSpa | 31.76 | 0.943 | 0.198 | 146 | 420 |
| | Ours(HQ) | 31.95 | 0.943 | 0.192 | 44 | 266 |
| | Ours(LM) | 31.51 | 0.937 | 0.205 | 28 | 227 |
| Counter | 3DGS | 29.09 | 0.906 | 0.201 | 393 | 893 |
| | Reduced3DGS | 28.57 | 0.899 | 0.212 | 67 | 331 |
| | GaussianSpa | 28.98 | 0.911 | 0.191 | 175 | 518 |
| | Ours(HQ) | 28.78 | 0.906 | 0.195 | 44 | 326 |
| | Ours(LM) | 28.35 | 0.892 | 0.220 | 25 | 272 |
| Kitchen | 3DGS | 31.24 | 0.925 | 0.126 | 581 | 1197 |
| | Reduced3DGS | 31.03 | 0.921 | 0.133 | 118 | 428 |
| | GaussianSpa | 31.50 | 0.929 | 0.127 | 163 | 443 |
| | Ours(HQ) | 31.51 | 0.927 | 0.127 | 51 | 309 |
| | Ours(LM) | 31.27 | 0.923 | 0.133 | 36 | 275 |
| Room | 3DGS | 31.55 | 0.918 | 0.220 | 474 | 1007 |
| | Reduced3DGS | 31.03 | 0.914 | 0.227 | 68 | 310 |
| | GaussianSpa | 31.61 | 0.922 | 0.211 | 141 | 401 |
| | Ours(HQ) | 31.56 | 0.923 | 0.207 | 36 | 250 |
| | Ours(LM) | 31.16 | 0.915 | 0.226 | 22 | 214 |

Table 6: **Tanks&Temples per scene results.** Best results are in red region , second best are in orange region . Memory (VRAM) values are in MB.

| Scene | Method | PSNR↑ | SSIM↑ | LPIPS↓ | Static VRAM↓ | Rendering VRAM↓ |
|---|---|---|---|---|---|---|
| Train | 3DGS | 21.97 | 0.818 | 0.198 | 397 | 773 |
| | Reduced3DGS | 21.74 | 0.804 | 0.219 | 74 | 227 |
| | GaussianSpa | 21.81 | 0.826 | 0.197 | 164 | 339 |
| | Ours(HQ) | 21.52 | 0.820 | 0.197 | 49 | 208 |
| | Ours(LM) | 21.32 | 0.818 | 0.205 | 35 | 160 |
| Truck | 3DGS | 25.39 | 0.881 | 0.143 | 752 | 1268 |
| | Reduced3DGS | 25.28 | 0.875 | 0.154 | 105 | 276 |
| | GaussianSpa | 25.65 | 0.887 | 0.126 | 115 | 448 |
| | Ours(HQ) | 25.37 | 0.886 | 0.121 | 54 | 214 |
| | Ours(LM) | 25.23 | 0.883 | 0.130 | 39 | 167 |

Table 7: **Mip-NeRF360 Outdoor per scene results.** Best results are in red region , second best are in orange region . Memory (VRAM) values are in MB.

| Scene | Method | PSNR↑ | SSIM↑ | LPIPS↓ | Static VRAM↓ | Rendering VRAM↓ |
|---|---|---|---|---|---|---|
| Bicycle | 3DGS | 25.20 | 0.764 | 0.210 | 1794 | 2955 |
| | Reduced3DGS | 25.10 | 0.763 | 0.219 | 314 | 686 |
| | GaussianSpa | 25.43 | 0.779 | 0.222 | 240 | 491 |
| | Ours(HQ) | 25.34 | 0.782 | 0.206 | 63 | 248 |
| | Ours(LM) | 25.16 | 0.771 | 0.226 | 48 | 205 |
| Stump | 3DGS | 26.61 | 0.769 | 0.217 | 1585 | 2487 |
| | Reduced3DGS | 26.75 | 0.778 | 0.218 | 318 | 663 |
| | GaussianSpa | 27.07 | 0.802 | 0.207 | 222 | 460 |
| | Ours(HQ) | 27.19 | 0.803 | 0.198 | 60 | 233 |
| | Ours(LM) | 27.04 | 0.797 | 0.213 | 47 | 198 |
| Garden | 3DGS | 27.36 | 0.863 | 0.108 | 1494 | 2449 |
| | Reduced3DGS | 27.14 | 0.858 | 0.116 | 303 | 684 |
| | GaussianSpa | 27.22 | 0.854 | 0.140 | 227 | 460 |
| | Ours(HQ) | 27.17 | 0.858 | 0.127 | 78 | 273 |
| | Ours(LM) | 26.62 | 0.846 | 0.143 | 63 | 240 |
| Flowers | 3DGS | 21.49 | 0.602 | 0.338 | 1066 | 1716 |
| | Reduced3DGS | 21.37 | 0.598 | 0.345 | 221 | 497 |
| | GaussianSpa | 21.60 | 0.624 | 0.332 | 209 | 451 |
| | Ours(HQ) | 21.65 | 0.625 | 0.331 | 66 | 255 |
| | Ours(LM) | 21.37 | 0.609 | 0.344 | 48 | 203 |
| Treehill | 3DGS | 22.58 | 0.633 | 0.328 | 1192 | 1947 |
| | Reduced3DGS | 22.41 | 0.630 | 0.337 | 241 | 563 |
| | GaussianSpa | 22.84 | 0.655 | 0.312 | 226 | 471 |
| | Ours(HQ) | 22.74 | 0.649 | 0.313 | 59 | 233 |
| | Ours(LM) | 22.44 | 0.640 | 0.335 | 43 | 188 |

Table 8: **DeepBlending per scene results.** Best results are in red region , second best are in orange region . Memory (VRAM) values are in MB.

| Scene | Method | PSNR↑ | SSIM↑ | LPIPS↓ | Static VRAM↓ | Rendering VRAM↓ |
|---|---|---|---|---|---|---|
| Playroom | 3DGS | 30.00 | 0.903 | 0.244 | 678 | 1197 |
| | Reduced3DGS | 29.96 | 0.904 | 0.248 | 90 | 276 |
| | GaussianSpa | 30.48 | 0.914 | 0.239 | 137 | 320 |
| | Ours(HQ) | 30.77 | 0.914 | 0.229 | 52 | 218 |
| | Ours(LM) | 30.58 | 0.912 | 0.243 | 30 | 167 |
| DrJohnson | 3DGS | 29.41 | 0.902 | 0.239 | 1129 | 1941 |
| | Reduced3DGS | 29.24 | 0.901 | 0.247 | 152 | 404 |
| | GaussianSpa | 29.50 | 0.910 | 0.239 | 184 | 424 |
| | Ours(HQ) | 29.56 | 0.909 | 0.237 | 56 | 269 |
| | Ours(LM) | 29.45 | 0.904 | 0.250 | 37 | 219 |

Table 9: **Real-time rendering performance comparison of different methods across various desktop and mobile devices in Bicycle scene.** All tests were conducted using a WebGL-based viewer, which we modified from the repository (Kwok & Ye, 2023) to support both Spherical Gaussians and 3rd-order Spherical Harmonics. Performance measured in FPS. "cannot render" indicates that the device's browser crashed or showed a black screen, while "render error" signifies incorrect rendering, such as color display issues. The best result is shown in **bold**.

| Method | RTX3060 | Dimensity 9400+ | Snapdragon 8+ Gen 1 | Snapdragon 888 |
|---|---|---|---|---|
| 3DGS | 26.3 | 6.6 | cannot render | connot render |
| GaussianSpa | **165.0** | 31.4 | render error | render error |
| Ours | **165.0** | **91.0** | **120.9** | **60.1** |

Table 10: **Storage Size Comparison.** We compare the storage size (MB) of our method against pruning-based methods (MaskGaussian, GaussianSpa) and the compact Gaussian approach (EAGLES). Our method not only surpasses pruning baselines but also achieves superior storage efficiency compared to EAGLES, even without employing explicit compression techniques.

| Method | Storage on MipNeRF 360 (MB) ↓ | Storage on DeepBlending (MB) ↓ |
|---|---|---|
| MaskGaussian | 271 | 156 |
| GaussianSpa | 115 | 104 |
| EAGLES | 54 | 52 |
| **Ours (HQ)** | 55 | 54 |
| **Ours (LM)** | 40 | 33 |

Table 11: **Quantitative comparison on the Shiny Blender Real dataset. Bold** figures indicate the best results.

| Method | PSNR ↑ | | | | SSIM ↑ | | | | LPIPS ↓ | | | |
|---|---|---|---|---|---|---|---|---|---|---|---|---|
| | Mean | Garden | Sedan | Toycar | Mean | Garden | Sedan | Toycar | Mean | Garden | Sedan | Toycar |
| Ref-NeRF | 23.62 | 22.01 | 25.21 | 23.65 | 0.646 | 0.584 | 0.720 | 0.633 | 0.264 | 0.251 | 0.234 | 0.231 |
| ENVIDR | 23.00 | 21.47 | 24.61 | 22.92 | 0.606 | 0.561 | 0.707 | 0.549 | 0.332 | 0.263 | 0.387 | 0.345 |
| 3DGS | 23.85 | 21.75 | 26.03 | 23.78 | 0.662 | 0.571 | 0.771 | 0.637 | **0.230** | 0.248 | 0.206 | 0.237 |
| 2DGS | 24.15 | 22.53 | 26.23 | 23.70 | 0.661 | 0.609 | 0.778 | 0.597 | 0.292 | 0.254 | 0.225 | 0.396 |
| GShader | 23.46 | 21.74 | 24.89 | 23.76 | 0.647 | 0.576 | 0.728 | 0.637 | 0.254 | 0.274 | 0.259 | 0.239 |
| R3DG | 21.98 | 21.92 | 21.18 | 22.83 | 0.619 | 0.556 | 0.643 | 0.657 | 0.349 | 0.354 | 0.380 | 0.312 |
| 3DGS-DR | 24.00 | 21.82 | 26.32 | 23.83 | 0.664 | 0.581 | 0.773 | 0.639 | **0.230** | **0.247** | 0.208 | **0.231** |
| Ref-Gaussian | **24.61** | 22.97 | **26.60** | 24.27 | 0.685 | 0.617 | 0.777 | 0.660 | 0.246 | 0.256 | 0.245 | 0.256 |
| **Ours** | 24.44 | **23.01** | 25.93 | **24.38** | **0.695** | **0.631** | **0.788** | **0.666** | **0.230** | 0.256 | **0.175** | 0.259 |

Table 12: **Comparison of training time across different datasets and hardware configurations.** We report the training duration (in minutes and seconds) for MEGS[2] on both RTX 3090 and RTX 4090 GPUs, and compare it with GaussianSpa and 3D Gaussian Splatting (3DGS) on an RTX 4090.

| Dataset | Scene | Ours | | GaussianSpa | 3DGS |
|---|---|---|---|---|---|
| | | RTX 3090 | RTX 4090 | RTX 4090 | RTX 4090 |
| **Mip-NeRF360** | Bicycle | 37min51s | 23min20s | 27min57s | 25min28s |
| | Stump | 34min09s | 22min34s | 26min18s | 20min48s |
| | Garden | 38min23s | 22min56s | 29min21s | 24min06s |
| | Flowers | 39min12s | 26min57s | 30min13s | 17min08s |
| | Treehill | 37min34s | 24min53s | 28min38s | 18min43s |
| | Bonsai | 51min49s | 28min19s | 44min14s | 16min36s |
| | Counter | 51min22s | 32min06s | 51min25s | 19min20s |
| | Kitchen | 50min33s | 33min17s | 49min36s | 22min59s |
| | Room | 41min05s | 28min53s | 44min41s | 20min23s |
| **Tanks&Temples** | Train | 28min39s | 19min08s | 28min45s | 11min06s |
| | Truck | 29min59s | 20min22s | 27min03s | 13min21s |
| **DeepBlending** | Playroom | 33min01s | 22min33s | 31min32s | 18min21s |
| | DrJohnson | 38min28s | 24min57s | 36min26s | 24min31s |

Table 13: **Comparison of the number of Gaussian primitives.** We report the total number of Gaussians (in millions) across three benchmark datasets.

| Method | Mip-NeRF360 | Tanks & Temples | DeepBlending |
|---|---|---|---|
| 3DGS | 2.718 | 1.568 | 2.461 |
| LP-3DGS | 1.866 | 1.116 | - |
| Mini-splatting | 0.559 | 0.320 | 0.397 |
| MaskGaussian | 1.205 | 0.590 | 0.694 |
| GaussianSpa | 0.528 | 0.447 | 0.409 |
| Ours(HQ) | 0.611 | 0.618 | 0.598 |
| Ours(LM) | 0.462 | 0.437 | 0.411 |

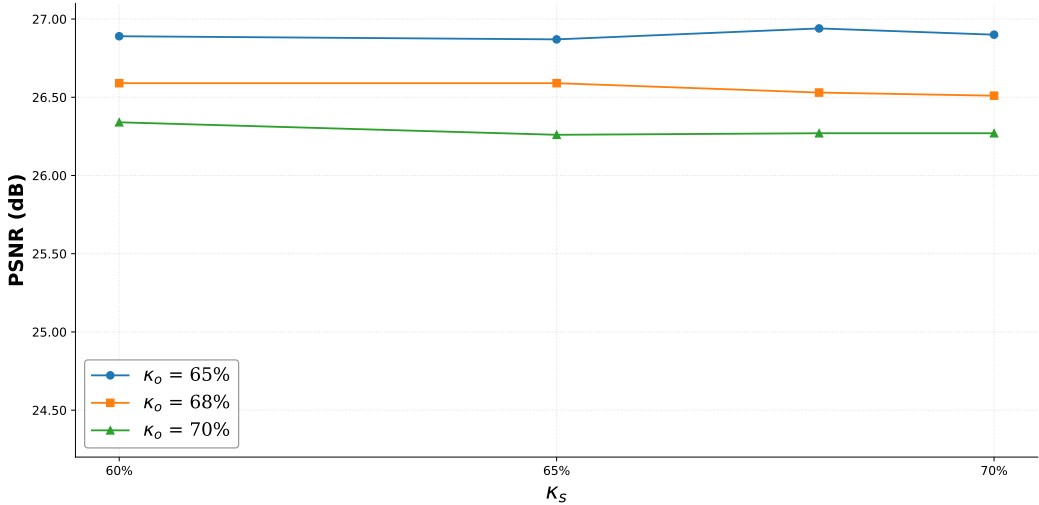

Figure 6: **Sensitivity analysis of pruning hyperparameters.** We investigate the impact of varying the lobe budget $\kappa_s$ (x-axis) and the primitive budget $\kappa_o$ (represented by different colored lines) on rendering quality (PSNR) and memory consumption on the garden scene. The values annotated in the boxes indicate the specific rendering VRAM usage for each configuration.

### A.4 MORE QUALITATIVE RESULTS

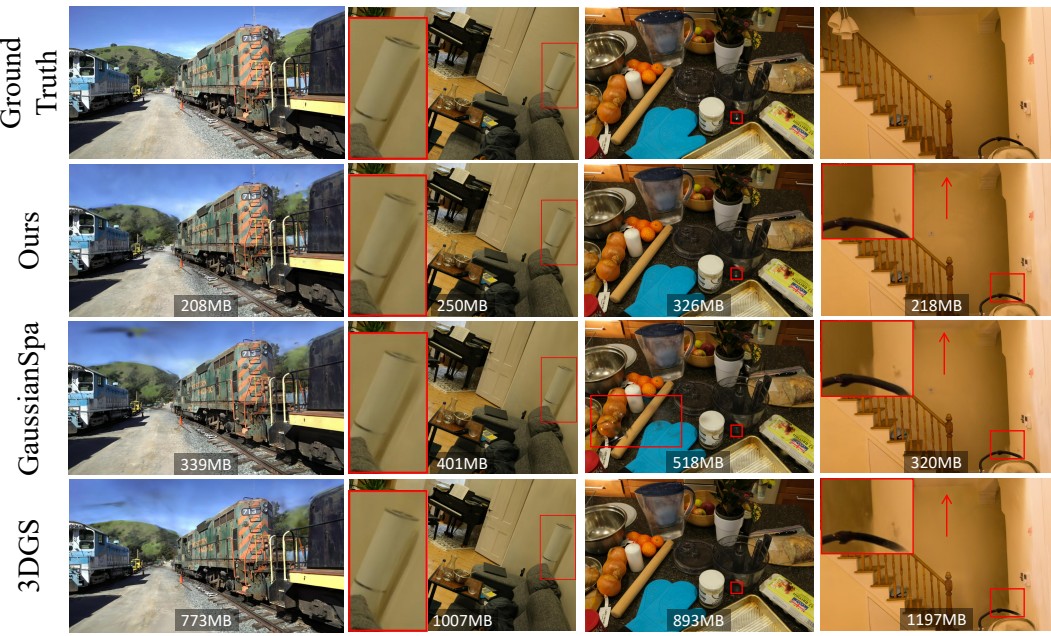

Figure 7: More qualitative results on Train, Room, Kitchen, Counter and Playroom scenes comparing to previous methods and the corresponding ground truth images. The rendering VRAM consumption for the corresponding method is annotated at the bottom of each image.

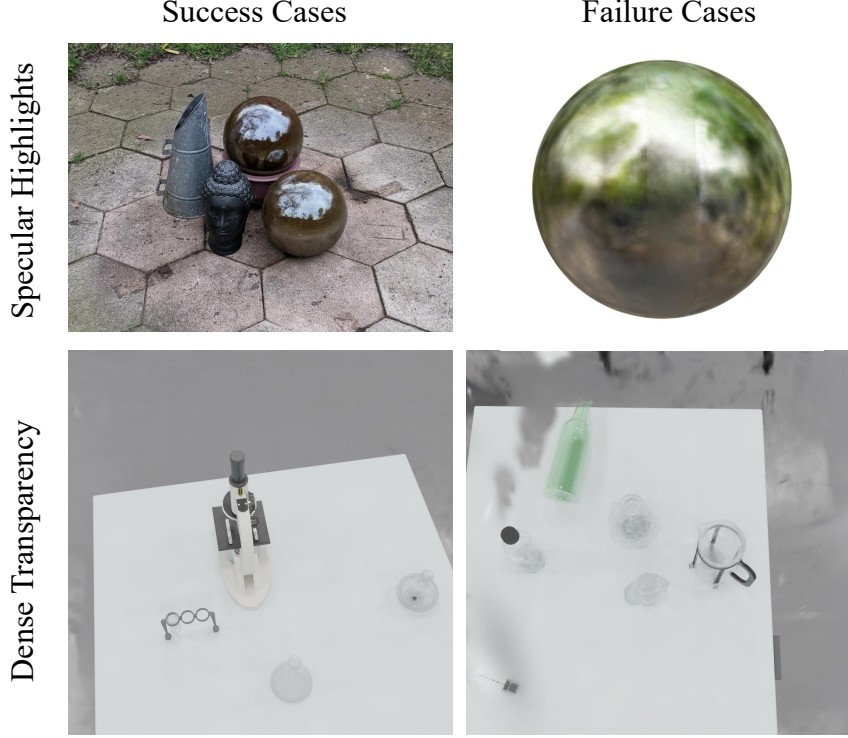

Figure 8: **Visualizations of success and failure cases under challenging material conditions.** The left column demonstrates successful reconstructions of scenes containing specular highlights and transparency. Conversely, the right column illustrates specific failure cases where we observe reconstruction artifacts under conditions of highly specular highlights or dense transparency.

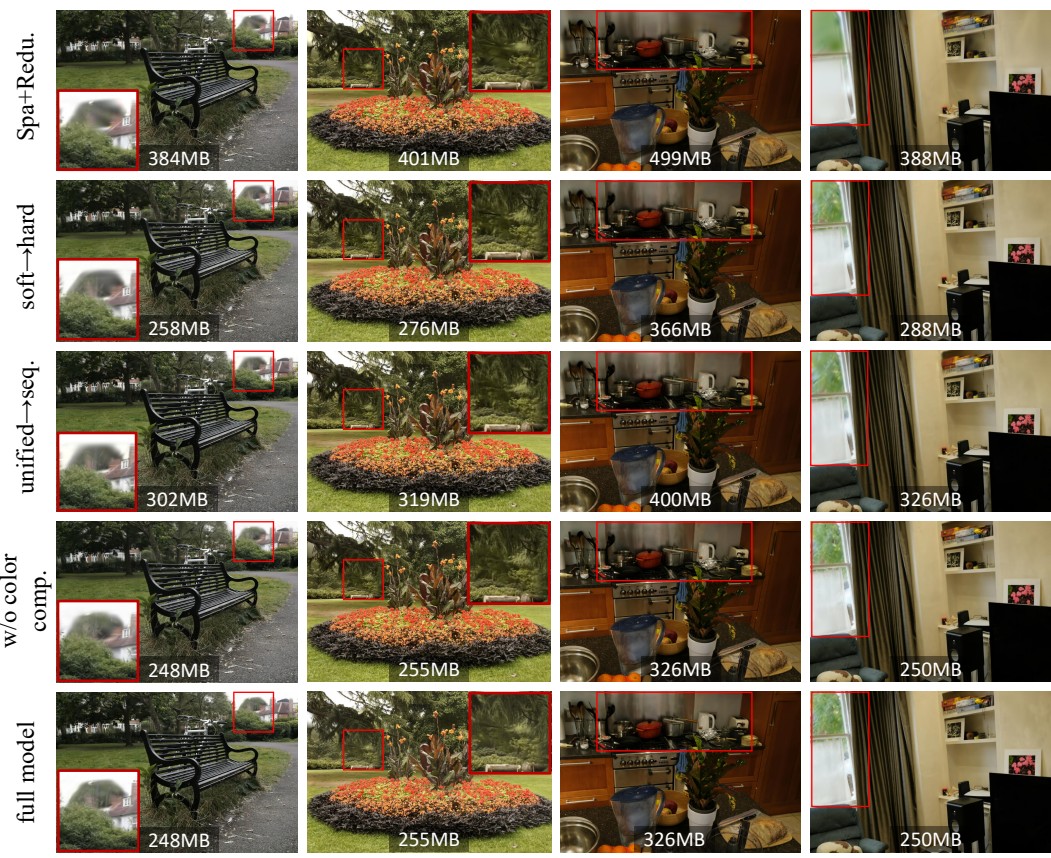

Figure 9: Qualitative ablation results on the Bicycle, Flowers, Counter and Room scenes. The rendering VRAM consumption for the corresponding method is annotated at the bottom of each image.

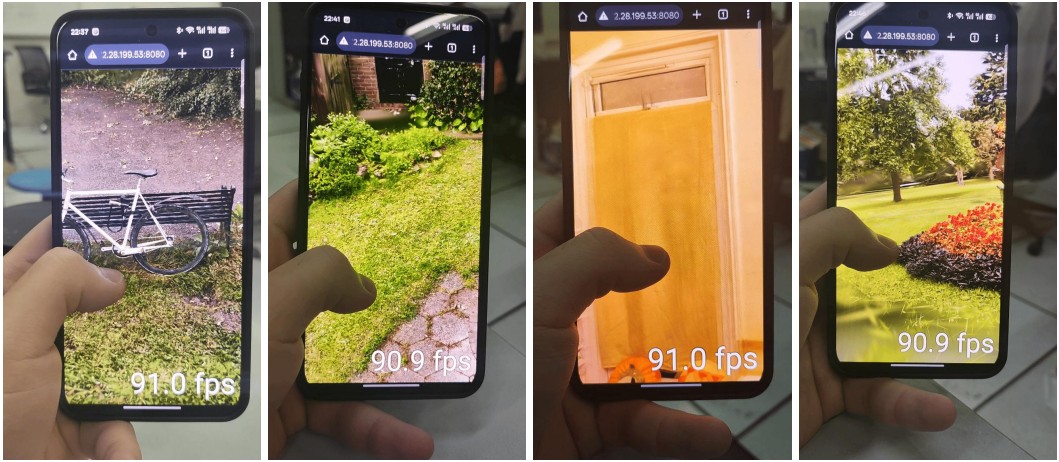

Figure 10: Real-time rendering performance of MEGS[2] on various scenes, showcased via a WebGL-based viewer on OnePlus Ace 5 Ultra with **MediaTek Dimensity 9400+**. The live frame rate (FPS) for each view is annotated in the imagery.

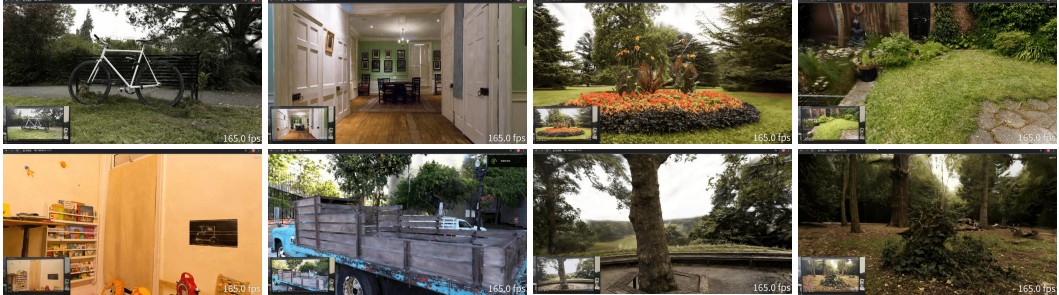

Figure 11: Real-time rendering performance of MEGS$^2$ on various scenes, showcased via a WebGL-based viewer on a lenovo laptop with **NVIDIA GeForce RTX3060 Laptop GPU**. The live frame rate (FPS) for each view is annotated in the imagery.

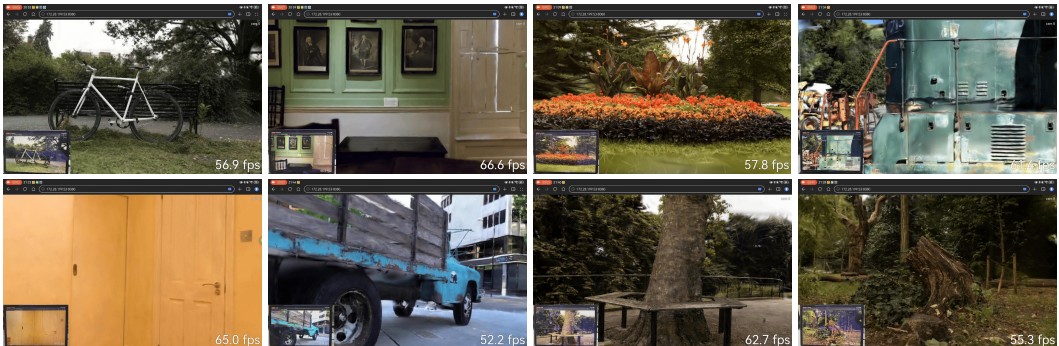

Figure 12: Real-time rendering performance of MEGS$^2$ on various scenes, showcased via a WebGL-based viewer on Huawei MatePad Air with **Qualcomm Snapdragon 888**. The live frame rate (FPS) for each view is annotated in the imagery.

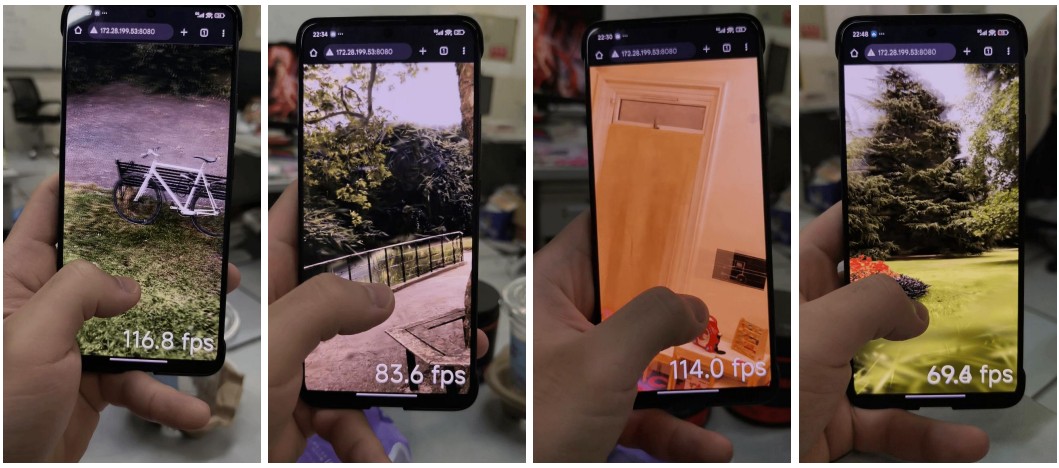

Figure 13: Real-time rendering performance of MEGS$^2$ on various scenes, showcased via a WebGL-based viewer on RedMi K60 with **Qualcomm Snapdragon 8+ Gen 1**. The live frame rate (FPS) for each view is annotated in the imagery.

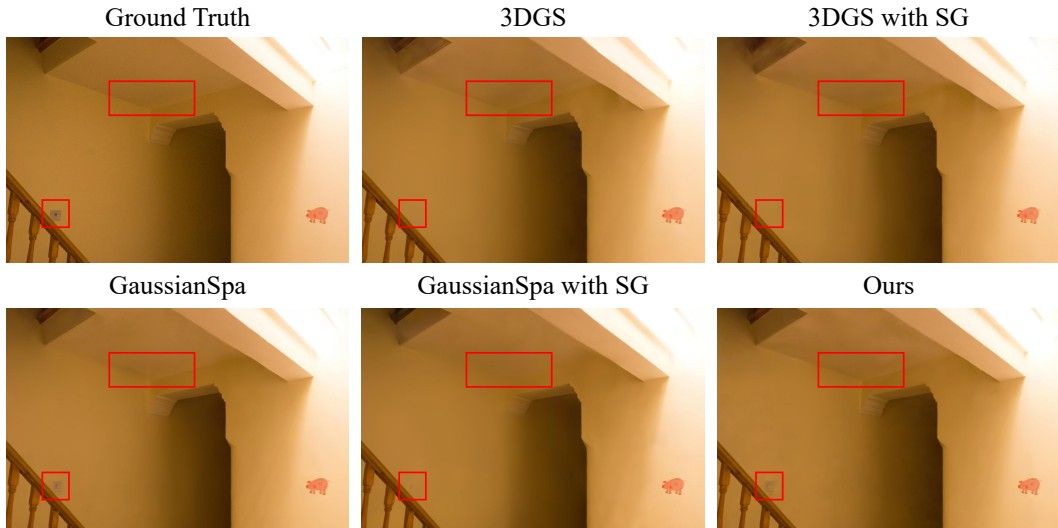

Figure 14: **Visual comparison in the Playroom scene.** "With SG" indicates the variant where Spherical Harmonics (SH) are replaced by Spherical Gaussians (SG). As observed in the figure, the speckled texture stems from the Ground Truth, which exhibits actual shadow regions alongside inherent image roughness that complicates training. While most methods smooth this detail out entirely, both GaussianSpa and MEGS² (Ours) manage a rough reconstruction.

## A.5 THE USE OF LARGE LANGUAGE MODELS (LLMS)

The article was refined using Gemini 2.5 Pro / Gemini 2.5 Flash / DeepSeek R1 to improve phrasing for natural English expression and correct grammatical errors.

