# OpenReview forum: "MEGS^{2}: Memory-Efficient Gaussian Splatting via Spherical Gaussians and Unified Pruning"
_ICLR.cc/2026/Conference — ICLR 2026 Poster_

### Official Review · Reviewer_vVWJ · 2025-10-18

**Soundness:** 3
**Presentation:** 3
**Contribution:** 2
**Rating:** 2
**Confidence:** 4

**Summary:**

This paper proposed a memory-efficient Gaussian Splatting method for edge devices. It uses spherical Gaussians to replace the original spherical harmonics for better efficiency. It also involves a pruning process to eliminate the redundant Gaussian points. Experimental results show the proposed method can reduce VRAM compared to the original 3DGS.

**Strengths:**

1. This paper is well-organized and easy to follow. Language is good.
2. The proposed method can render comparable results to 3DGS.
3. The presentation is good. The demonstrated effect is impressive.

**Weaknesses:**

1. In the abstract, "Experiments show that MEGS2 achieves a 50% static VRAM reduction and a 40% rendering VRAM reduction ...", this sentence confuses me. What's static VRAM and rendering VRAM? There's no clear claim.

2. In lines 60-62, the authors claim, "However, these methods require decoding the full Gaussian parameters from a compressed
state before rendering, even resulting in a larger rendering memory than the 3DGS methods without compression.".  I am wondering if the proposed method doesn't need to decode the compressed Gaussian parameters for rendering? I did not find a clear description.

3. The authors claim the proposed spherical Gaussian is their contribution. However, the spherical Gaussian was already proposed by [1]. The authors do not present a clear discussion of it. I did not find a detailed description for arbitrarily-oriented SG. How to achieve it? Why is it better than orthogonal? Is there any theoretical analysis?

4. The proposed memory-constraint optimization is too straightforward. Please elaborate on the contribution. The formulation did not claim the relation between opacity $o$ and sharpness $s$.

5. To the proposed post-processing procedure, the authors claim they have a three-step strategy. However, I only find two: (1) removal according to low opacity, (2) by sharpness. Moreover, I do not think it is novel. It is also straightforward. There's no specific design in it.

6. As for experiments, the authors do not compare with state-of-the-art GS compression methods, like LocoGS[2], speedy-splat[3], LightGaussian[4], maskGaussian[5].


7. There's no visual comparison for the ablation study. I think it would be better to present it. There are no Gaussian numer comparisons.
It would be better to show the training costs and storage memory. There are no detailed VRAM analyses for each Gaussian attribute.






[1] Wang Y, Chen S, Yi R. Sg-splatting: Accelerating 3d gaussian splatting with spherical gaussians[J]. arXiv preprint arXiv:2501.00342, 2024.

[2] Shin S, Park J, Cho S. Locality-aware gaussian compression for fast and high-quality rendering[J]. arXiv preprint arXiv:2501.05757, 2025.

[3] Hanson A, Tu A, Lin G, et al. Speedy-splat: Fast 3d gaussian splatting with sparse pixels and sparse primitives[C]//Proceedings of the Computer Vision and Pattern Recognition Conference. 2025: 21537-21546.


[4] Fan Z, Wang K, Wen K, et al. Lightgaussian: Unbounded 3d gaussian compression with 15x reduction and 200+ fps[J]. Advances in neural information processing systems, 2024, 37: 140138-140158.

[5] Liu Y, Zhong Z, Zhan Y, et al. Maskgaussian: Adaptive 3d gaussian representation from probabilistic masks[C]//Proceedings of the Computer Vision and Pattern Recognition Conference. 2025: 681-690.

**Questions:**

No code attached. I think it would be better if the authors released the code.

Have the authors try 4DGS? In my opinion, it is quite easy to adapt the proposed method for 4DGS.

---

> ### Author Response · Authors · 2025-11-23
>
> ### **Rebuttal to Reviewer vVWJ**
>
> We sincerely thank the reviewer for acknowledging the impressive effect of our method and for providing constructive feedback, particularly regarding the clarity of our contribution.
>
> ---
>
> **W1: What's static VRAM and rendering VRAM?**
>
> **R1:** We provided the definitions for static VRAM and rendering VRAM in the "Evaluation Details" paragraph of Appendix A.2:
>
> * Static VRAM refers to the VRAM required to load all 3DGS primitives into the renderer and dequantize or decode them into a ready-to-render state. This is not necessarily equal to the file size, as some methods might represent Gaussian attributes in a more compact or lower-precision data format but still require restoring them to full, 32-bit precision Gaussian attributes before rendering.
> * Rendering VRAM denotes the peak VRAM usage during the rendering process, which is typically larger than the static VRAM due to the introduction of intermediate variables (e.g., projected 2D Gaussian attributes, key-value tables for tile-based rendering) during rendering.
>
> We also detailed the specific measurement method for VRAM in the same paragraph.
>
> ---
>
> **W2: I am wondering if the proposed method doesn't need to decode the compressed Gaussian parameters for rendering?**
>
> **R2:** Yes, our method does not require a decoding process. This is because our approach relies solely on pruning (both on primitive-wise and lobe-wise) and does not depend on common compression schemes that use neural networks or hash grids. Gaussian primitives derived by our method possess a varying number of spherical Gaussian lobes and can be directly rendered without any decoding process.
>
> ---
>
> **W3: The authors claim the proposed spherical Gaussian is their contribution. However, the spherical Gaussian was already proposed by \[1\]. The authors do not present a clear discussion of it.**
>
> **R3:** Our key contribution is not simply replacing SHs with SGs, but that we prune the redundant SGs inside each remaining Gaussian primitive, besides pruning the redundant Gaussian primitives. We find that pruning both Gaussian primitives and SGs inside each retained Gaussian primitive simultaneously leads to a strongly memory-efficient scene representation, which is not explored by previous works like SG-Splatting. Furthermore, to achieve this pruning, we design a more flexible SG representation with an arbitrary orientation. In contrast, SG-Splatting uses fixed axes in SG, and SG-Splatting mixes SHs with SGs, resulting in a heavier and more complex representation with a larger memory consumption.
>
> ---
>
> **W4: I did not find a detailed description for arbitrarily-oriented SG. How to achieve it? Why is it better than orthogonal? Is there any theoretical analysis?**
>
> **R4:** The formulation is detailed in Section 3.2 (Eq. 1 & 2), and the advantages are discussed in the "Choice of arbitrarily-oriented SG lobes" paragraph. As noted in R3, our design enables lobe-wise pruning, which is infeasible for orthogonal SGs (a.k.a fixed-axis SGs) due to their rigid global axis alignment, and offers superior expressivity. Theoretically, since orthogonal SGs are merely a constrained special case of our formulation, our arbitrarily oriented model strictly subsumes their solution space, guaranteeing better representational power.
>
> ---
>
> **W5: The proposed memory-constraint optimization is too straightforward. Please elaborate on the contribution. The formulation did not claim the relation between opacity and sharpness.**
>
> **R5:** Our core contribution lies in generalizing the optimization objective from simply minimizing the number of primitives to minimizing the **total memory consumption**. We formally define this objective and solve it by simultaneously pruning redundant Gaussian primitives and the internal parameters (SG lobes) within each retained Gaussian, which is not explored in prior work.
>
> Opacity is an attribute of the Gaussian primitive, whereas sharpness descripes the contribution of each spherical lobe to a Gaussian primitive. In Eq. 3, the $L\_0$ norm of opacity represents the number of effective Gaussian primitives, while the $L\_0$ norm of sharpness represents the number of effective spherical lobes (see Sec. 3.3.2). By weighting these terms with their memory costs, we formulate the total memory consumption. Thus, no explicit functional relation exists between them; they are independent variables coupled solely by the global VRAM constraint.
>
> ---

---

> ### Author Response · Authors · 2025-11-23
>
> **W6: To the proposed post-processing procedure, the authors claim they have a three-step strategy. However, I only find two: (1) removal according to low opacity, (2) by sharpness. Moreover, I do not think it is novel. It is also straightforward. There's no specific design in it.**
>
> **R6:** Our key contribution lies in pruning both the Gaussian primitives and parameters (SG lobe number) inside each Gaussian primitives. Our post-processing procedure has three key steps as stated in Section 3.3.3:
>
> 1. Removal of Gaussian Primitives according to low opacity.
> 2. Removal of Spherical Gaussian lobes according to low sharpness with color compensation.
> 3. Finetuning to recover rendering quality.
>
> ---
>
> **W7: As for experiments, the authors do not compare with state-of-the-art GS compression methods, like LocoGS[2], speedy-splat[3], LightGaussian[4], maskGaussian[5].**
>
> **R7:** Thanks for raising these related works. Our experiments in Table 1 already cover extensive SOTA methods like Mini-Splatting, GaussianSpa, EAGLES and MesonGS. Following your suggestion, we have further integrated LightGaussian, LocoGS, and MaskGaussian into the revised manuscript and due to limited time, we will further add speedy splat in the further revision.  . These comparisons confirm that our method remains SOTA in VRAM consumption, achieving rendering quality comparable to or slightly exceeding these baselines.
>
> ---
>
> **W8: There's no visual comparison for the ablation study. I think it would be better to present it.**
>
> **R8:** We have incorporated visual comparisons for the ablation study in the revised manuscript (Figure 9). These results demonstrate that our pruning strategy effectively reduces VRAM consumption while maintaining, or even enhancing, rendering quality.
>
> ---
>
> **W9: There are no Gaussian number comparisons. It would be better to show the training costs and storage memory.**
>
> **R9:** We appreciate the suggestions and have listed the number of Gaussians for our method and other pruning-based approches in the table below,  which has been incorporated into the revised manuscript. The results show that our Gaussian count remains highly competitive, even though our primary objective is minimizing total VRAM.
>
> | Method | Mip-NeRF360 | Tank & Temples | DeepBlending |
> | :---- | :---: | :---: | :---: |
> | 3DGS | 2.718 | 1.568 | 2.461 |
> | LP-3DGS | 1.866 | 1.116 | \- |
> | Mini-splatting | 0.559 | 0.320 | 0.397 |
> | MaskGaussian | 1.205 | 0.590 | 0.694 |
> | GaussianSpa | 0.528 | 0.447 | 0.409 |
> | Ours(HQ) | 0.611 | 0.618 | 0.598 |
> | Ours(LM) | 0.462 | 0.437 | 0.411 |
>
> Moreover, our method not only prunes the primitive count but also reduces the internal parameters per Gaussian (via lobe-wise pruning), as demonstrated in Table 3 and Figure 5\.
>
> We have also included the reported training costs(Table 12 in Appendix) and storage memory(Table 10 in Appendix) in the revised paper.
>
> ---
>
> **W10:** **There are no detailed VRAM analyses for each Gaussian attribute.**
>
> **R10:** Regarding the VRAM analysis, we provided a breakdown in the **Introduction**: "Specifically, the rendering VRAM can be divided into a static component and a dynamic component. The static part relates directly to the total number of Gaussian primitives loaded into the renderer, which is a product of the primitive count and the parameters per primitive. On the other hand, the dynamic part is also related to the primitive count in the specific camera viewpoint. This inherent relationship underscores that to reduce the overall memory footprint, we need to reduce both the number of primitives and the per-primitive parameter size." The details of VRAM composition and measurement are in Appendix A.2 ("Evaluation Details"), and Appendix A.3 Table 3 compares the number of parameters used for color modeling across different methods.
>
> ---
>
> **Q1: No code attached. I think it would be better if the authors released the code.**
>
> **R11:** We plan to open-source all our code upon publication, including our improved WebGL renderer, evaluation scripts, and training scripts.
>
> ---
>
> **Q2: Have the authors try 4DGS? In my opinion, it is quite easy to adapt the proposed method for 4DGS.**
>
> **R12:** Our core operations, specifically primitive pruning, SG color modeling and SG lobe-wise pruning, are inherently compatible with 4DGS frameworks and offer potential for direct adaptation. However, fully maximizing compression necessitates addressing temporal redundancy. Formally incorporating such temporal constraints is non-trivial. Therefore, we consider this extension out of the scope of our current work and leave it for future research.

---

### Official Review · Reviewer_RnYG · 2025-10-30

**Soundness:** 3
**Presentation:** 3
**Contribution:** 3
**Rating:** 4
**Confidence:** 4

**Summary:**

This paper presents a memory-efficient Gaussian Splatting framework that replaces high-dimensional spherical harmonics with lightweight spherical Gaussian lobes. It also introduces a unified pruning strategy for both Gaussian primitives and spherical Gaussians, resulting in improved pruning performance due to its regularization effect. Experimental results demonstrate that the proposed framework achieves substantial reductions in both static and dynamic VRAM usage while providing superior rendering quality.

**Strengths:**

- The paper effectively analyzes the key factors of large VRAM consumption and mitigates them using the proposed techniques.

- The use of spherical Gaussian verifies to be an efficient alternative for color representation, achieving meaningful parameter reduction while maintaining a comparable representation capacity to spherical harmonics.

- Experimental results show that the proposed method outperforms various 3DGS baselines in rendering quality while requiring only minimal VRAM consumption.

**Weaknesses:**

- The lack of technical novelty needs to be enhanced. The main performance gain for the memory reduction originates from the replacement of spherical harmonics with spherical Gaussian lobes. However, spherical Gaussians have already been applied to 3DGS in prior work, such as SG-Splatting.

- Pruning solely based on the opacity value may lead to performance limitations, since the contribution of Gaussians is also influenced by other attributes, such as scales.

- Despite the low VRAM consumption, this representation requires a relatively high storage demand compared to compact Gaussian approaches.

**Questions:**

- Could you compare the proposed pruning method with other pruning metrics from existing approaches, such as CompactGaussian [CVPR'24], EAGLES [ECCV'24], and LightGaussian [NeurIPS'24]?

-  Could a lower degree of spherical harmonics be more efficient than spherical Gaussian lobes? It could be helpful to compare the diverse SH degrees against the proposed representation.

---

> ### Author Response · Authors · 2025-11-23
>
> ### **Rebuttal to Reviewer RnYG**
>
> We sincerely thank the reviewer for the positive evaluation and for recognizing our key strengths in mitigating VRAM consumption, employing efficient Spherical Gaussians, and achieving a superior quality-memory trade-off.
>
>
>
> ------
>
>
>
> **W1: The main performance gain for the memory reduction originates from the replacement of spherical harmonics with spherical Gaussian lobes. However, spherical Gaussians have already been applied to 3DGS in prior work, such as SG-Splatting.**
>
> **R1:** Our key contribution is not simply replacing SHs with SGs, but that we prune the redundant SGs inside each remaining Gaussian primitive, besides pruning the redundant Gaussian primitives. We find that pruning both Gaussian primitives and SGs inside each retained Gaussian primitive simultaneously leads to a strongly memory-efficient scene representation, which is not explored by previous works like SG-Splatting. Furthermore, to achieve this pruning, we design a more flexible SG representation with an arbitrary orientation. In contrast, SG-Splatting uses fixed axes in SG, and SG-Splatting mixes SHs with SGs, resulting in a heavier and more complex representation with a larger memory consumption.
>
>
>
> ------
>
>
>
> **W2: Pruning solely based on the opacity value may lead to performance limitations, since the contribution of Gaussians is also influenced by other attributes, such as scales.**
>
> **R2:** We clarify that while attributes like scale are implicitly integrated into the optimization process. Though our memory constraints count the number of active Gaussian primitives, which is represented by $L_0$ norm of opacity, we set the Gaussian primitive to be inactive (opacity=0) if they do not contribute enough in the rendering. Computing such a contribution involves attributes like scales.
>
> Specifically, our constraint formulation (Eq. 3) includes only opacity and sharpness because their $L_0$ norms directly quantify VRAM consumption (primitive and lobe counts), whereas other attributes do not define the memory budget. Crucially, the optimization process—specifically the projection step (Eqs. 7-9)—does incorporate other attributes. As detailed in Appendix A.2, our Importance-Based Selection calculates pruning scores where attributes like scales implicitly influence the decision (via Eq. A.17). We have verified that this strategy outperforms a pure opacity-based approach in outdoor scenes, as shown below:
>
> | **Operator / Scene**        | **PSNR $\\uparrow$** | **SSIM $\\uparrow$** | **LPIPS $\\downarrow$** | **VRAM $\\downarrow$ (MB)** |
> | --------------------------- | -------------------- | -------------------- | ----------------------- | --------------------------- |
> | **Indoor Scene (bonsai)**   |                      |                      |                         |                             |
> | Magnitude-Based             | 31.29                | 0.930                | 0.216                   | 273MB                       |
> | Importance-Based            | **31.95**            | **0.943**            | **0.192**               | **266MB**                   |
> | **Outdoor Scene (bicycle)** |                      |                      |                         |                             |
> | Magnitude-Based             | 24.61                | 0.734                | 0.262                   | 259MB                       |
> | Importance-Based            | **25.34**            | **0.782**            | **0.206**               | **248MB**                   |
>
>
>
> ------
>
>
>
> **W3: Despite the low VRAM consumption, this representation requires a relatively high storage demand compared to compact Gaussian approaches.**
>
> **R3:** No, we don’t trade the storage for memory. Our method inherently yields a compact representation, with uncompressed storage typically ranging between 30–40 MB. Moreover, it is fully compatible with standard compression techniques like quantization or vector quantization. For instance, applying simple FP16 quantization reduces the storage size to approximately 10 MB on the Room scene, which is comparable to those storage compression methods but our method uses much less runtime memory.
>
>
>
> ------

---

> ### Author Response · Authors · 2025-11-23
>
> **Q1: Could you compare the proposed pruning method with other pruning metrics from existing approaches, such as CompactGaussian [CVPR'24], EAGLES [ECCV'24], and LightGaussian [NeurIPS'24]?**
>
> **R4:** Table 1 already includes comparisons with CompactGaussian [CVPR'24] and EAGLES [ECCV'24]. We have supplemented Table 1 with a comparison against LightGaussian [NeurIPS'24] and more baselines in the revised manuscript. These comparisons confirm that our method remains SOTA in rendering VRAM consumption, achieving rendering quality comparable to or slightly exceeding these baselines.
>
>
>
> ------
>
>
>
> **Q2: Could a lower degree of spherical harmonics be more efficient than spherical Gaussian lobes? It could be helpful to compare the diverse SH degrees against the proposed representation.**
>
> **R5:** In Section 3.2 ("Choice of arbitrarily-oriented SG lobes") and Table 3 (Appendix), we have shown that our SG design, across different parameter budgets, consistently outperforms Spherical Harmonics (SH) of diverse degrees. This demonstrates that SG lobes are a more efficient representation than low-degree SH for view-dependent color modeling. We will further emphasize this in our revision to address your concerns.

---

### Official Review · Reviewer_FDsv · 2025-10-31

**Soundness:** 4
**Presentation:** 3
**Contribution:** 3
**Rating:** 8
**Confidence:** 4

**Summary:**

The paper presents MEGS2, a memory-efficient framework for 3D Gaussian Splatting (3DGS) that targets the rendering VRAM bottleneck rather than only the storage footprint. The authors propose two key ideas: (1) replacing high-order Spherical Harmonics (SH) color modeling with Spherical Gaussians (SG) for compact, local view-dependent representation, and (2) a unified soft pruning framework that simultaneously prunes redundant Gaussian primitives and per-primitive SG lobes under a total memory constraint, formulated as a constrained optimization problem solved by an ADMM-inspired method. Experiments show the effectiveness of the proposed method. The approach is further validated on both desktop and mobile devices, showing strong frame rates in a WebGL viewer.

**Strengths:**

- Replacing SH with arbitrarily oriented Spherical Gaussians is an effective solution to reduce per-primitive parameters. The formulation is physically intuitive and well integrated into the 3DGS framework.
- The paper provides detailed quantitative and qualitative results across multiple datasets and baselines.
- The WebGL-based real-time rendering results demonstrate real usability on mobile platforms.
- The method’s ability to generalize across devices and maintain reasonable quality is a clear step toward deployable Gaussian-based rendering.

**Weaknesses:**

- Several important parameters are not explicitly described or justified. The choice of these hyperparameters may significantly affect performance and reproducibility.
- Scenes with highly specular highlights or dense transparency are acknowledged as future work but should be illustrated. The method’s performance on dynamic or large-scale scenes remains unknown.
- While the authors promise code release, the current text lacks explicit training schedules, optimization hyperparameters, and hardware runtimes, which hinders replication.
- The convergence and optimality properties of the ADMM-inspired procedure are not formally analyzed.

**Questions:**

See weaknesses.

---

> ### Author Response · Authors · 2025-11-23
>
> ### **Rebuttal to Reviewer FDsv**
>
> We sincerely thank the reviewer for recognizing our arbitrarily oriented SG design as a physically intuitive solution, while also acknowledging our method's real-world usability on mobile platforms via WebGL.
>
>
> ------
>
>
>
> **W1: Several important parameters are not explicitly described or justified. The choice of these hyperparameters may significantly affect performance and reproducibility.**
>
> **R1:** We thank you for this crucial reminder. In Appendix A.2 (Implementation Details), we stated that most parameters followed the original 3DGS and GaussianSpa implementations. We will revise the manuscript to explicitly list the specific values for all key optimization hyperparameters. Furthermore, we will include a sensitivity analysis on the bonsai scene showing how rendering quality is affected by key parameters, as demonstrated in the table below:
>
> | **primitive budget $\kappa_o$** | **lobe budget $\kappa_s$** | **PSNR** |
> | ------------------------------- | -------------------------- | -------- |
> | 0.65                            | 0.68                       | 32.08    |
> | 0.65                            | 0.7                        | 31.88    |
> | 0.65                            | 0.72                       | 31.85    |
> | 0.65                            | 0.75                       | 31.83    |
> | 0.65                            | 0.78                       | 31.76    |
> | 0.68                            | 0.75                       | 31.74    |
> | 0.7                             | 0.75                       | 31.47    |
>
>
>
> ------

---

> ### Author Response · Authors · 2025-11-23
>
> **W2: Scenes with highly specular highlights or dense transparency are acknowledged as future work but should be illustrated.**
>
> **R2:** Our method generally handles reflective objects effectively, despite challenges in extreme specular or transparent scenarios. Existing benchmarks like MipNeRF 360 already include such attributes, where our model demonstrates strong performance (see Figure 3 and Appendix Tables 5-8). We further validated this on the Shiny Blender Real dataset, achieving results comparable to specialized methods like Ref-Gaussian (see table below).
>
> | **Method**   | **PSNR $\uparrow$ (Mean)** | **PSNR (gardensphere)** | **PSNR (sedan)** | **PSNR (toycar)** | **SSIM $\uparrow$ (Mean)** | **SSIM (gardensphere)** | **SSIM (sedan)** | **SSIM (toycar)** | **LPIPS $\downarrow$ (Mean)** | **LPIPS (gardensphere)** | **LPIPS (sedan)** | **LPIPS (toycar)** |
> | ------------ | -------------------------- | ----------------------- | ---------------- | ----------------- | -------------------------- | ----------------------- | ---------------- | ----------------- | ----------------------------- | ------------------------ | ----------------- | ------------------ |
> | Ref-NeRF     | 23.62                      | 22.01                   | 25.21            | 23.65             | 0.646                      | 0.584                   | 0.720            | 0.633             | 0.264                         | 0.251                    | 0.234             | 0.231              |
> | ENVIDR       | 23.00                      | 21.47                   | 24.61            | 22.92             | 0.606                      | 0.561                   | 0.707            | 0.549             | 0.332                         | 0.263                    | 0.387             | 0.345              |
> | 3DGS         | 23.85                      | 21.75                   | 26.03            | 23.78             | 0.662                      | 0.571                   | 0.771            | 0.637             | **0.230**                     | 0.248                    | 0.206             | 0.237              |
> | 2DGS         | 24.15                      | 22.53                   | 26.23            | 23.70             | 0.661                      | 0.609                   | 0.778            | 0.597             | 0.292                         | 0.254                    | 0.225             | 0.396              |
> | GShader      | 23.46                      | 21.74                   | 24.89            | 23.76             | 0.647                      | 0.576                   | 0.728            | 0.637             | 0.254                         | 0.274                    | 0.259             | 0.239              |
> | R3DG         | 21.98                      | 21.92                   | 21.18            | 22.83             | 0.619                      | 0.556                   | 0.643            | 0.657             | 0.349                         | 0.354                    | 0.380             | 0.312              |
> | 3DGS-DR      | 24.00                      | 21.82                   | 26.32            | 23.83             | 0.664                      | 0.581                   | 0.773            | 0.639             | **0.230**                     | **0.247**                | 0.208             | **0.231**          |
> | Ref-Gaussian | **24.61**                  | 22.97                   | **26.60**        | 24.27             | 0.685                      | 0.617                   | 0.777            | 0.660             | 0.246                         | 0.256                    | 0.245             | 0.256              |
> | Ours         | 24.44                      | **23.01**               | 25.93            | **24.38**         | **0.695**                  | **0.631**               | **0.788**        | **0.666**         | **0.230**                     | 0.256                    | **0.175**         | 0.259              |
>
> However, we acknowledge artifacts in highly specular or dense transparent cases; visualizations of these failure cases have been added to Appendix Figure 8. Addressing these complex scenarios likely requires integrating the full rendering equation, which we consider a promising direction for future work.
>
> ------
>
> **W3:** **The method’s performance on dynamic scenes remains unknown.**
>
> **R3:** Our core operations, specifically primitive pruning, SG color modeling and SG lobe-wise pruning, are inherently compatible with 4DGS frameworks and offer potential for direct adaptation. However, fully maximizing compression necessitates addressing temporal redundancy. Formally incorporating such temporal constraints is non-trivial. Therefore, we consider this extension out of the scope of our current work and leave it for future research.
>
> ------

---

> ### Author Response · Authors · 2025-11-23
>
> **W4: The method’s performance on large-scale scenes remains unknown.**
>
> **R4:** As for large-scale scenes, our experiments cover a variety of environments (13 scenes in total), including several large-scale scenes like Train and Truck from the Tanks & Temples dataset, and various outdoor scenes from MipNeRF 360. Appendix Tables 5-8 provide per-scene metrics to demonstrate the robustness of our method across these environments. For qualitative results, please refer to Figure 3 and Appendix Figure 8.
>
>
>
> ------
>
>
>
> **W5: While the authors promise code release, the current text lacks explicit training schedules, optimization hyperparameters, and hardware runtimes, which hinders replication.**
>
> **R5:** We apologize for the lack of clarity on these details.
>
> - Training Schedules: We confirm that Appendix A.2 (Implementation Details) provided the specific strategy: "Regarding the training strategy, we employ the same densification, pruning, and fine-tuning procedures as GaussianSpa, conducting densification training, sparse training, and fine-tuning for the same number of steps."
> - Optimization Hyperparameters: As noted in R1, we will explicitly list all critical parameter values in the revised manuscript to aid replication.
> - Hardware Runtimes: While our Table 4 and Table 9 already contain hardware runtimes in the form of Frame Rates (FPS), we agree that the Training Cost (Hardware Runtime) needs to be included for completeness. We will supplement the manuscript with Table 12 showing the training time for our method across key datasets:
>
> | **Dataset**       | **Ours(RTX 3090)** | **Ours(RTX 4090)** | **GaussianSpa (RTX 4090)** | **3DGS (RTX 4090)** |
> | ----------------- | ------------------ | ------------------ | -------------------------- | ------------------- |
> | **Mip-NeRF360**   | 42min26s           | 27min01s           | 36min55s                   | 20min36s            |
> | **Tanks&Temples** | 29min19s           | 19min45s           | 27min54s                   | 12min13s            |
> | **DeepBlending**  | 35min44s           | 23min45s           | 33min59s                   | 21min26s            |
>
>
>
> ------
>
>
>
> **W6: The convergence and optimality properties of the ADMM-inspired procedure are not formally analyzed.**
>
> **R6:** We thank the reviewer for raising this crucial theoretical point.
>
> We acknowledge that providing rigorous formal analysis for global optimality and convergence is challenging because our optimization problem involves non-convex objectives and non-convex constraints (stemming from the sparsity-inducing norms on opacity and sharpness), and we utilize a modern optimizer (Adam) rather than simple gradient descent. This context makes strict theoretical guarantees difficult to achieve, which is common in applied non-convex sparse optimization.
>
> However, the ADMM-inspired procedure was chosen for its empirical effectiveness in decoupling the complex constraints. For quantitative verification, please refer to '8. Additional Discussion' and Figure 9 in the GaussianSpa paper (Zhang et al., 2025b).

---

### Official Review · Reviewer_1h2z · 2025-11-01

**Soundness:** 3
**Presentation:** 3
**Contribution:** 2
**Rating:** 6
**Confidence:** 3

**Summary:**

The paper presents MEGS^2, a 3D Gaussian Splatting framework that cuts both primitive count and per-primitive parameters to fix the rendering VRAM bottleneck.
The paper replaces SH colors with lightweight, arbitrarily oriented spherical Gaussians (SG) so most Gaussians use only a few lobes.
The paper’s method formulates primitive and lobe pruning as one memory-constrained optimization, reaching ~8× static and ~6× rendering VRAM compression over vanilla 3DGS, still close to SOTA quality.

The paper backs its claims with extensive experiments on standard datasets (MipNeRF 360, etc), with qualitative evaluations.
Ablation studies show different pruning scheme is outperformed by the method presented by the paper, unified soft pruning.

**Strengths:**

The paper provides following strenghts:
- Unlike storage-only methods, it reduces static and rendering VRAM (50% static, 40% rendering vs SOTA), so it runs on edge/mobile.
- Usage of SG color model is prunable. SGs keep high-freq/view-dependent details with about half the params of SH and allow lobe-wise pruning.
- Unified soft pruning works better than sequential. ADMM-style optimization over opacity and sharpness finds a better quality–memory trade-off than “prune primitives then prune SH.”

**Weaknesses:**

The paper suffers from following weaknesses:
- Optimization pipeline is complex. Needs proxy variables, proximal projection, then post-processing + fine-tuning, so it’s harder to integrate than simple importance-pruning.
- As stated in the limitation section, the paper mostly focuses on renderer-agnostic static memory. Paper explicitly leaves renderer-specific dynamic VRAM optimization “for future work,” so some engines may still hit a ceiling.
- As authors noted, performance on “highly complex highlights” needs more study, so scenes with extreme view-dependent effects may degrade first.

**Questions:**

- The authors argue SGs are “more amenable to pruning.” Can the authors quantify how many lobes are actually used per Gaussian at convergence, per scene?
- The optimization looks more complex than sequential pruning: how sensitive is it to the pruning budget? Do small changes lead to big quality swings?

---

> ### Author Response · Authors · 2025-11-23
>
> ### **Rebuttal to Reviewer 1h2z**
>
> We sincerely thank the reviewer for their insightful feedback and for accurately recognizing our core contributions regarding VRAM reduction and the prunable SG design.
>
> ---
>
> **W1: Optimization pipeline is complex. Needs proxy variables, proximal projection, then post-processing \+ fine-tuning, so it’s harder to integrate than simple importance-pruning.**
>
> **R1:** We acknowledge that our pipeline is relatively complex, yet every design choice has been empirically validated and is non-redundant (Table 2 confirms this). Compared to simpler importance-pruning methods (e.g., Mini-Splatting), which can reduce memory by about half (e.g., $477 \text{ MB} \rightarrow 265 \text{ MB}$ in Table 1\) while slightly improving rendering quality, we firmly believe the complexity is justified by the performance gains.
>
> Regarding integrability, our code implementation is quite straightforward: it requires adding only a few steps within the main training loop (primarily updating proxy/dual variables and updating the loss). The post-processing stage is comparable in implementation complexity to other pruning methods, and the fine-tuning stage is simply a continuation of the training loop.
>
> ---
>
> **W2: As stated in the limitation section, the paper mostly focuses on renderer-agnostic static memory. Paper explicitly leaves renderer-specific dynamic VRAM optimization “for future work,” so some engines may still hit a ceiling.**
>
> **R2:**  Our method focuses on pruning the number of Gaussians and the parameters of each remaining Gaussian, so our method reduces both the static memory (storing the whole scene) and the dynamic rendering memory (temporal memory consumption when rendering from a specific viewpoint). Our limitation means that the dynamic rendering memory could be further reduced using CUDA-level optimization, which is not explored in our paper. However, we believe our method is compatible with these optimizations. Hope this could address your concern about this limitation.
>
> ---
>
> **Q1: The authors argue SGs are "more amenable to pruning." Can the authors quantify how many lobes are actually used per Gaussian at convergence, per scene?**
>
> **R3:** Appendix Table 3 demonstrates that our lobe-wise pruning reduces the average color parameters from 24 to 9.7 on the DeepBlending dataset, which is significantly lower than the 48 parameters in the original 3DGS. The average lobe usage and the distribution of Gaussians with different lobe counts are visualized in Figure 5, and we provide the specific numerical values in the table below:
>
> | Scene | Initial State (L=3, P=24) | Final Lobes ($L$) | Final Parameters ($P$) |
> | :---- | :---: | :---: | :---: |
> | Bicycle | (3, 24\) | 1.183 | 11.28 |
> | Stump | (3, 24\) | 1.328 | 12.30 |
> | Garden | (3, 24\) | 1.426 | 13.00 |
> | Flowers | (3, 24\) | 1.183 | 11.28 |
> | Treehill | (3, 24\) | 1.225 | 11.58 |
> | Bonsai | (3, 24\) | 1.116 | 10.81 |
> | Counter | (3, 24\) | 1.224 | 11.57 |
> | Kitchen | (3, 24\) | 1.267 | 11.87 |
> | Room | (3, 24\) | 1.072 | 10.50 |
>
> ---
>
> **Q2: The optimization looks more complex than sequential pruning: how sensitive is it to the pruning budget? Do small changes lead to big quality swings?**
>
> **R4:** Rendering quality is insensitive to the pruning budget and does not exhibit significant swings. Please refer to the table below for quantitative results on the bicycle scene; we have also added this experiment as Figure 6 in the Appendix.
>
> | Pruning Budget ($\kappa$) | Config( $\kappa_o$ $\kappa_s$) | PSNR |
> | :---- | :---: | :---: |
> | 280MB | 0.65 0.6 | 25.39 |
> | 270MB | 0.65 0.68 | 25.34 |
> | 260MB | 0.68 0.6 | 25.37 |
> | 250MB | 0.7 0.6 | 25.27 |

---

> ### Author Response · Authors · 2025-12-02
> **Supplementary Response to Reviewer 1h2z: Weakness 3**
>
> **W3: As authors noted, performance on “highly complex highlights” needs more study, so scenes with extreme view-dependent effects may degrade first.**
>
> **R5:** Regarding the potential performance degradation in scenes with "highly complex highlights," we have conducted additional experiments on the Shiny Blender Real dataset. As shown in Table 11, MEGS² achieves performance comparable to specialized methods (e.g., Ref-Gaussian), demonstrating strong robustness even under extreme view-dependent effects.
> | **Method**   | **PSNR $\uparrow$ (Mean)** | **PSNR (gardensphere)** | **PSNR (sedan)** | **PSNR (toycar)** | **SSIM $\uparrow$ (Mean)** | **SSIM (gardensphere)** | **SSIM (sedan)** | **SSIM (toycar)** | **LPIPS $\downarrow$ (Mean)** | **LPIPS (gardensphere)** | **LPIPS (sedan)** | **LPIPS (toycar)** |
> | ------------ | -------------------------- | ----------------------- | ---------------- | ----------------- | -------------------------- | ----------------------- | ---------------- | ----------------- | ----------------------------- | ------------------------ | ----------------- | ------------------ |
> | Ref-NeRF     | 23.62                      | 22.01                   | 25.21            | 23.65             | 0.646                      | 0.584                   | 0.720            | 0.633             | 0.264                         | 0.251                    | 0.234             | 0.231              |
> | ENVIDR       | 23.00                      | 21.47                   | 24.61            | 22.92             | 0.606                      | 0.561                   | 0.707            | 0.549             | 0.332                         | 0.263                    | 0.387             | 0.345              |
> | 3DGS         | 23.85                      | 21.75                   | 26.03            | 23.78             | 0.662                      | 0.571                   | 0.771            | 0.637             | **0.230**                     | 0.248                    | 0.206             | 0.237              |
> | 2DGS         | 24.15                      | 22.53                   | 26.23            | 23.70             | 0.661                      | 0.609                   | 0.778            | 0.597             | 0.292                         | 0.254                    | 0.225             | 0.396              |
> | GShader      | 23.46                      | 21.74                   | 24.89            | 23.76             | 0.647                      | 0.576                   | 0.728            | 0.637             | 0.254                         | 0.274                    | 0.259             | 0.239              |
> | R3DG         | 21.98                      | 21.92                   | 21.18            | 22.83             | 0.619                      | 0.556                   | 0.643            | 0.657             | 0.349                         | 0.354                    | 0.380             | 0.312              |
> | 3DGS-DR      | 24.00                      | 21.82                   | 26.32            | 23.83             | 0.664                      | 0.581                   | 0.773            | 0.639             | **0.230**                     | **0.247**                | 0.208             | **0.231**          |
> | Ref-Gaussian | **24.61**                  | 22.97                   | **26.60**        | 24.27             | 0.685                      | 0.617                   | 0.777            | 0.660             | 0.246                         | 0.256                    | 0.245             | 0.256              |
> | Ours         | 24.44                      | **23.01**               | 25.93            | **24.38**         | **0.695**                  | **0.631**               | **0.788**        | **0.666**         | **0.230**                     | 0.256                    | **0.175**         | 0.259              |

---

### Official Review · Reviewer_zwHD · 2025-11-02

**Soundness:** 3
**Presentation:** 1
**Contribution:** 1
**Rating:** 2
**Confidence:** 5

**Summary:**

This paper proposes memory efficient gaussian splatting method by combining Spherical Gaussian and ADMM-inspired unified pruning. Noticed from related paper SG-Splatting, they found fixed orthogonal axes of SG lobes limit rendering performance, and they make it arbitrary oriented. Inspired from GuassianSpa, they optimize proxy of opacity and sharpness parameters simulatenously to sparsify the values. At the end of optimization of proxies, they prune primitives based on significance of the proxies. They found sequential pruning (prune gaussians first then prune parameters) shows poorer trade-off between primitive count and primitive parameters than unified pruning (prune gaussians and parameters together). To recover rendering quality degradation caused by pruning, they further finetune diffuse color based on color loss. Their experiment results show comparable rendering quality of their method against state-of-the-art method but significant save in VRAM, tested on standard benchmarking datasets.

**Strengths:**

This paper addresses the limited color representation capacity of SG-Splatting and further introduces opacity- and sharpness-based pruning, along with energy recovery of removed lobes, to reduce VRAM usage with maintaining high reconstruction quality.

**Weaknesses:**

I think the major issue is the inaccurate contribution claim of this paper. As discussed in Questions 1 and 5, the first contribution claim seems exaggerated, and their findings are not well supported by ablation experiments. Similarly, the second contribution claim is also inaccurate, as discussed in Question 2. Moreover, poor presentation of the method, such as missing information on model choices, makes it difficult for readers to fully understand their approach. Please refer to my Questions for detailed comments.

Presentation issue:
* I could not find the authors’ conclusion regarding model selection.
  1. In lines 306–311, the authors propose two distinct operators: **sharpness-based selection** and **range-based selection**. Which one is ultimately adopted in the final model and why? I could not find a statement clarifying this choice.
  2. Similarly, in lines 785–787, which criterion was finally chosen and why?: **magnitude-based selection** or **importance-based selection**?
* I could not identify the setting difference between the last two rows in Table 1. Please provide distinct method names (e.g., ours-setting_A and ours-setting_B) and explain them clearly in the appropriate section of the main text.
* Please include a citation for ADMM at line 234.

**Questions:**

* The first contribution claim in lines 97–99 makes it sound as if the authors are the first to propose substituting Gaussian Splats with Spherical Gaussians, which is not true, as this was already introduced by SG-Splatting. The authors should instead highlight the distinction between **fixed axes** and **arbitrarily oriented axes** in Spherical Gaussians. Emphasizing the reduction of per-primitive parameter count and VRAM savings is also inappropriate, since these advantages originate from SG-Splatting.
* The second contribution claim in lines 100–102 also seems inaccurate. The memory-constrained problem and its solution were originally introduced by GaussianSpa, and the authors simply extend that method by incorporating the sharpness of Gaussians to retain those with better expressive capacity. The authors should revise the statement of contribution 2 to accurately reflect their original findings.
* The VRAM reduction of the proposed method is primarily attributed to the smaller number of parameters used for encoding view-specific color and pruning. To provide supporting evidence for the pruning aspect, I suggest that the authors present both VRAM consumption and the number of Gaussians in Table 1. This will help readers better understand the pruning performance compared to pruning-only baselines such as LP-3DGS, Mini-Splatting, and GaussianSpa.
* The qualitative results in Figure 3 do not fully support the authors’ claim that "MEGS² delivers a cleaner and more complete reconstruction in Playroom" (line 415). When examining the ceiling area in the top center of the rendered images, MEGS² produces a somewhat speckled texture, whereas GaussianSpa and 3DGS yield cleaner textures. Additionally, the light switch in the bottom-left area appears blurrier in MEGS² compared to GaussianSpa.
* I am wondering about the effect of using arbitrarily oriented Spherical Gaussians. Please include an ablation study to analyze this effect. For example, the authors could test:
  1. The proposed method with **Spherical Harmonics (SH)**
  2. The proposed method with **fixed-axis Spherical Gaussians**
* Furthermore, I am curious why **GaussianSpa (SH → SG)** shows a quality drop (27.01dB in Table 2) compared to the original GaussianSpa (27.56dB in Table 1). This raises doubts about the justification for using Spherical Gaussians.

---

> ### Author Response · Authors · 2025-11-23
>
> ### **Rebuttal to Reviewer zwHD**
>
> We sincerely thank you for your valuable and highly constructive feedback. We acknowledge that there are ambiguities in the presentation of our current manuscript that may have led to misunderstandings. We appreciate this opportunity to clarify our contributions and address your concerns.
>
> ---
>
> **Q1: The first contribution claim in lines 97–99 makes it sound as if the authors are the first to propose substituting Gaussian Splats with Spherical Gaussians, which is not true, as this was already introduced by SG-Splatting.**
>
> **R1:**  Our key contribution is not simply replacing SHs with SGs, but that we prune the redundant SGs inside each remaining Gaussian primitive, besides pruning the redundant Gaussian primitives. We find that pruning both Gaussian primitives and SGs inside each retained Gaussian primitive simultaneously leads to a strongly memory-efficient scene representation, which is not explored by previous works like SG-Splatting. Furthermore, to achieve this pruning, we design a more flexible SG representation with an arbitrary orientation. In contrast, SG-Splatting uses fixed axes in SG, and SG-Splatting mixes SHs with SGs, resulting in a heavier and more complex representation with a larger memory consumption.
>
> ---
>
> **Q2: The memory-constrained problem and its solution were originally introduced by GaussianSpa.**
>
> **R2:**  The fundamental difference lies in the formulation: while GaussianSpa addresses pruning solely by reducing the number of Gaussian primitives, we formulate the problem as minimizing the total memory consumption, which not only prunes the number of Gaussians but also the parameter (SG lobes) inside each retained Gaussian, which is not explored by previous works.  We demonstrate in our experiments that our method leads to a much more memory-efficient representation with comparable rendering quality as previous methods.
>
> **Q3: The authors simply extend that method by incorporating the sharpness of Gaussians.**
> We would like to clarify the role of 'sharpness': It is an attribute of the *SG lobes*, not the Gaussian primitive. It is not used to filter important Gaussians. As detailed in Sec 3.3.3, a sharpness of 0 implies no view-dependent effect, allowing that lobe to be merged into the base color. Thus, the $L\_0$ norm of the flatten sharpness vector formalizes the  *total number of active lobes*.
>
> Our core novelty is formulating a unified constrained optimization problem (Eq. 3\) that minimizes *both* primitive count (via opacity) and lobe count (via sharpness) under a single VRAM budget.
>
> ---
>
> **Q4: I suggest that the authors present both VRAM consumption and the number of Gaussians in Table 1\.**
>
> **R4:** We appreciate this suggestion and have listed the number of Gaussians for our method and other pruning-based approaches in the table below,  which has been incorporated into the revised manuscript.
>
> | Method | Mip-NeRF360 | Tank & Temples | DeepBlending |
> | :---- | :---: | :---: | :---: |
> | 3DGS | 2.718 | 1.568 | 2.461 |
> | LP-3DGS | 1.866 | 1.116 | \- |
> | Mini-splatting | 0.559 | **0.320** | **0.397** |
> | MaskGaussian | 1.205 | 0.590 | 0.694 |
> | GaussianSpa | 0.528 | 0.447 | 0.409 |
> | Ours(HQ) | 0.611 | 0.618 | 0.598 |
> | Ours(LM) | **0.462** | 0.437 | 0.411 |
>
> Crucially, our VRAM reduction stems from compressing both the number of Gaussians and the internal parameters per Gaussian (via lobe-wise pruning), as further detailed in Table 3 and Figure 5\.
>
> ---
>
> **Q5: The qualitative results in Figure 3 do not fully support the authors’ claim.**
>
> **R5:** We acknowledge that GaussianSpa yields smoother textures on the ceiling and light switch. However, MEGS² delivers a cleaner reconstruction in other regions, specifically the wall corner where GaussianSpa struggles Our claim is that we have comparable performance or slightly superior overall quantitative metrics (e.g., Playroom PSNR 30.77 dB vs. 30.48 dB) than GaussianSpa, but we maintain this competitive fidelity while consuming roughly 50% less rendering VRAM (182MB vs. 370MB).
>
> ---

---

> ### Author Response · Authors · 2025-11-23
>
> **Q6: I am wondering about the effect of using arbitrarily oriented Spherical Gaussians. Please include an ablation study to analyze this effect.**
>
> **R6:** We thank the reviewer for this question. The effectiveness of our arbitrarily-oriented SGs is primarily demonstrated in Figure 4 . This figure explicitly compares our design against SH and fixed-axis SGs ('Ortho-SG') at various parameter budgets, showing our design is far more expressive and efficient.
>
> Regarding the suggested ablations:
>
> 1. "Our method \+ SH": SH is inherently less amenable to pruning compared to SGs. Unlike our SG lobes, which possess identical parameter counts and computational patterns, SH coefficients vary significantly across orders in both parameter size and calculation rules. This heterogeneity makes SH unsuitable for our unified $L\_0$ pruning formulation. The closest experiment is 'Spa \+ Redu.' in Table 2, which is unstable and performs poorly.
>
> 2. "Our method \+ fixed-axis SGs": As discussed in R1, the fixed-axis SG design inherently does not support lobe-wise pruning. Since the axes are rigidly fixed to the global x/y/z directions, reducing the lobe count would restrict specular highlights solely to these specific axes, which is physically unrealistic.
>
> ---
>
> **Q7: Furthermore, I am curious why GaussianSpa (SH → SG) shows a quality drop (27.01dB in Table 2\) compared to the original GaussianSpa (27.56dB in Table 1). This raises doubts about the justification for using Spherical Gaussians.**
>
> **R7:** We believe this may be because GaussianSpa's parameters and proximal operators are likely better tuned for SH. A naive swap to SGs, without adapting the pruning heuristics, naturally leads to this suboptimal result. The fact that our full MEGS² framework recovers this performance (Table 2\) suggests that our unified pruning mechanism provides a beneficial regularization effect. This result does not imply SGs are inferior for color modeling; as we demonstrate in Figure 4, our SG design is highly expressive and efficient.
>
> ---
>
> **W1: As discussed in Questions 1 and 5, the first contribution claim seems exaggerated... Similarly, the second contribution claim is also inaccurate, as discussed in Question 2\.**
>
> **R8:** We hope our detailed clarifications in R1, R2, R3, and R6, along with the supporting evidence from our tables and figures, have successfully addressed these concerns and clarified the novelty and validity of our contributions.
>
> ---
>
> **W2: I could not find the authors’ conclusion regarding model selection.**
> **In lines 306–311, the authors propose two distinct operators: sharpness-based selection and range-based selection. Which one is ultimately adopted in the final model and why? …**
>
> **R9:**  Thanks for pointing out this issue and we would clarify more about this implementation details.  In our final model, we use range-based selection for sharpness (as it slightly outperforms the simpler sharpness-based selection). Regarding primitive pruning, while GaussianSpa recommends a hybrid strategy (i.e., magnitude-based for indoor scenes and importance-based for outdoor scenes), we opted for a unified approach in our experiments for simplicity and consistently utilized Importance-Based Selection across all datasets. The comparison below justifies this choice, showing that Importance-Based Selection is crucial for outdoor scenes while remaining robust for indoor ones.
>
> | Operator / Scene | PSNR $\uparrow$ | SSIM $\uparrow$ | LPIPS $\downarrow$ | VRAM $\downarrow$ (MB) |
> | :---- | :---: | :---: | :---: | :---: |
> | **Indoor Scene (bonsai)** |  |  |  |  |
> | Magnitude-Based | 31.29 | 0.930 | 0.216 | 273MB |
> | Importance-Based | **31.95** | **0.943** | **0.192** | **266MB** |
> | **Outdoor Scene (bicycle)** |  |  |  |  |
> | Magnitude-Based | 24.61 | 0.734 | 0.262 | 259MB |
> | Importance-Based | **25.34** | **0.782** | **0.206** | **248MB** |
>
> ---
>
> **W3: I could not identify the setting difference between the last two rows in Table 1\. Please provide distinct method names (e.g., ours-setting\_A and ours-setting\_B) and explain them clearly in the appropriate section of the main text.**
>
> **R10:** The two rows for our method in Table 1 represent a "High-Quality" (HQ) setting and a "Low-Memory" (LM) setting, focused on quality and compression, respectively. Because we could trade the memory for the performance, we show these two results. We have explicitly named these in the revised table following your suggestion.
>
> ---
>
> **W4: Please include a citation for ADMM at line 234\.**
> **R11:**  We have added the proper citation for ADMM at line 234\.

---

> > ### Comment · Reviewer_zwHD · 2025-11-24
> > **I appreciate authors detailed response**
> >
> > Thank the authors for your elaborate and detailed explanation. Your effort addressing the abmguities now leads me better understand of your work. I have follow-up comments and questions below.
> >
> > **Q1**. As reviewer **vVWJ** raised a point (W3) similar to mine, the per-primitive parameter count and VRAM reduction are outcomes of using Spherical Gaussians (SG), regardless of whether the lobe axes are fixed.
> > According to the SG-Splatting paper, SGs require only 10 parameters compared to the 48 parameters used by Spherical Harmonics (c.f., the last paragraph of the “SG-based Color Representation” section in SG-Splatting paper), and they report nearly a 50% reduction in VRAM usage (c.f., the Mip-NeRF360 column in Table 3 of SG-Splatting paper).
> > To the best of my knowledge, the advancement introduced by MEGS² over SG-Splatting is the enhancement of SG’s color representation by allowing the orthogonal axes to vary freely, which improves color expressivity and also enables pruning of SG lobes.
> >
> > "Our key contribution is not simply replacing SHs with SGs, but that we prune the redundant SGs inside each remaining Gaussian primitive, besides pruning the redundant Gaussian primitives. We find that pruning both Gaussian primitives and SGs inside each retained Gaussian primitive simultaneously leads to a strongly memory-efficient scene representation, which is not explored by previous works like SG-Splatting." -> I acknowledge that splat and SG-lobe pruning is the authors’ original and key contribution. However, this corresponds to their second claim (lines 100–102) and cannot simultaneously serve as the first claim.
> > Therefore, I suggested that the authors revise their first claim to highlight the distinction between fixed-axis and arbitrarily oriented Spherical Gaussians.
> >
> > **Q2 and Q3**. Thank you for the nuanced explanation. I think the reformulation of GaussianSpa from the perspective of memory consumption is a smart idea. These questions are now clear to me.
> >
> > **Q4**. Thank you for the additional analysis. The newly provided information indicates that on the DeepBlending dataset, MEGS² achieves a slightly lower pruning ratio than Mini-splatting while using less memory, which implies that the proposed pruning strategy can save additional memory (through lobe pruning) while also improving rendering quality.
> > However, when comparing GaussianSpa and Ours (HQ) on the Mip-NeRF360 dataset, GaussianSpa performs more aggressive pruning and still attains better PSNR metrics. Based on these two observations, it seems to me that the effectiveness of the proposed pruning method in maintaining quality is scene-dependent rather than universal.
> > I am wondering how the authors view the scene-dependent performance of their pruning method.
> >
> > **Q5**. I acknowledge that MEGS² achieves better fidelity than GaussianSpa according to the PSNR metrics. However, I am still curious about the cause of the speckled or blurry artifacts observed in MEGS². Are these artifacts due to the proposed pruning strategy, or are they inherent limitations of SG? I would appreciate it if the authors could provide additional discussion on this matter.
> >
> > **Q6, Q7, W1, W2, W3, and W4**. Thank you for the clarification. All of my questions have been resolved.

---

> > > ### Author Response · Authors · 2025-11-27
> > >
> > > Thank you for the positive feedback and for confirming that our explanation has clarified the previous ambiguities. We appreciate the continued discussion and address your follow-up comments below.
> > >
> > > **Q1: To the best of my knowledge, the advancement introduced by MEGS² over SG-Splatting is the enhancement of SG’s color representation by allowing the orthogonal axes to vary freely, which improves color expressivity and also enables pruning of SG lobes. … Therefore, I suggested that the authors revise their first claim to highlight the distinction between fixed-axis and arbitrarily oriented Spherical Gaussians.**
> > >
> > > **R1:** We thank the reviewer for the summary. As you noted, the advantages of our SG design over the fixed-axis SG in SG-Splatting primarily lie in superior expressivity and prunability.
> > >
> > > Following your suggestion, we will refine our contribution statement to shift the focus from simply "replacing SH with SG" to explicitly emphasizing the **"arbitrarily oriented"** and **"prunable"** characteristics, ensuring our novelty is highlighted more precisely. We greatly appreciate your advice.
> > >
> > > ---
> > >
> > > **Q2:** (Q2, Q3) Thank you for the nuanced explanation. I think the reformulation of GaussianSpa from the perspective of memory consumption is a smart idea. These questions are now clear to me.
> > >
> > > **R2:** We sincerely thank the reviewer for the positive feedback. We are encouraged that you recognize the value of our memory-consumption formulation and are glad that these questions are now resolved.
> > >
> > > ---
> > >
> > > **Q3: The newly provided information indicates that ... Based on these two observations, it seems to me that the effectiveness of the proposed pruning method in maintaining quality is scene-dependent rather than universal. I am wondering how the authors view the scene-dependent performance of their pruning method.**
> > >
> > > **R3:** Regarding the hypothesis of "scene-dependent" performance, we would like to share the following perspective:
> > >
> > > 1. Our strategy, combining primitive and lobe-wise pruning, maintains rendering quality and significantly compresses VRAM across datasets without exhibiting obvious scene dependency. As noted in Table 13, while Ours (HQ) retains slightly more primitives than GaussianSpa, our total VRAM remains significantly lower (\~40% less) due to lobe-wise pruning, all while achieving comparable PSNR and superior LPIPS. This pattern—slightly higher primitive count, significantly lower total VRAM, and comparable quality—is highly consistent on the DeepBlending dataset as well.
> > > 2. Detailed results in Tables 5-8 further confirm this consistency. Across individual scenes, our quality metrics remain comparable to GaussianSpa (with slight variations favoring either method, especially on LPIPS while the relative VRAM reduction consistently hovers around 40-50%. Overall, our method demonstrates stable and consistent performance across diverse environments.
> > >
> > > ---
> > >
> > > **Q4: However, I am still curious about the cause of the speckled or blurry artifacts observed in MEGS². Are these artifacts due to the proposed pruning strategy, or are they inherent limitations of SG?**
> > >
> > > **R4:** We have carefully investigated the source of these artifacts by cross-comparing the Ground Truth (GT), 3DGS, 3DGS with SG, GaussianSpa with SG, GaussianSpa, and MEGS²(Appendix, Figure 14).
> > >
> > > We found the speckled texture stems from the GT, which exhibits actual shadow regions alongside inherent image roughness that complicates training.
> > >
> > > Regarding the light switch, while most methods smooth this detail out entirely, both GaussianSpa and MEGS² manage a rough reconstruction. We acknowledge GaussianSpa resolves this specific object slightly better, however, MEGS² achieves this competitive fidelity while consuming 50% less rendering VRAM, validating our superior efficiency-quality trade-off.
> > >
> > > ---
> > >
> > > **Q5: (Q6, Q7, W1, W2, W3, and W4) Thank you for the clarification. All of my questions have been resolved.**
> > >
> > > **R5:** We thank the reviewer for their time and are pleased that our clarifications have successfully resolved all these concerns.

---

> > > > ### Comment · Reviewer_zwHD · 2025-11-27
> > > >
> > > > Thanks for the authors addressing my concerns. I am satisfying all other questions but Q1.
> > > >
> > > > **Q1**: Thanks for revising the statement for Contribution 1. However, my confusion remains unresolved. I may have missed an important point in our discussion about Q1. Could the authors justify the sentence *"which significantly reduces the per-primitive parameter count and thus lowers rendering VRAM with minimal impact on quality"* in lines 98-99? What I am pointing out is that this claim stems from substituting the color representation model from Spherical Harmonics to Spherical Gaussians, which is originally introduced in the SG-Splatting paper. If the authors are arguing that this advantage is instead achieved by simultaneously pruning Gaussians and lobes after modifying Spherical Harmonics to make them prunable, then this appears to duplicate the claim made in Contribution 2.

---

> > > > > ### Author Response · Authors · 2025-11-28
> > > > >
> > > > > Thank you for the follow-up. We agree with your assessment that the concept of using Spherical Gaussians (SG) for color representation on 3DGS  was initially explored by the SG-Splatting paper.
> > > > >
> > > > > We understand the concern regarding the conceptual similarity, and we would like to gently highlight that the key distinction of our method lies in successfully making this substitution standalone and quality-preserving, which distinguishes it from SG-Splatting and our own Contribution 2:
> > > > >
> > > > > 1. Hybrid vs. Standalone (Full Substitution): SG-Splatting employs a hybrid model (SH + SG) rather than achieving a full substitution like ours. As evidenced by their Table 2, their color modeling still relies heavily on SHs because the fixed-axis SG (a.k.a Multi-Axis Orthogonal SG) alone performs poorly. This is why we maintain that validating SGs as a standalone color modeling tool (by replacing SH completely with minimal impact on quality) is a distinct and significant contribution of our work, separate from the pruning contribution.
> > > > >
> > > > > 2. Memory Efficiency: SG-Splatting's Table 3 reports a storage (NOT VRAM) compression of ~50%, but the actual rendering VRAM reduction is only 20%-37%. As stated in our previous response: "In contrast, SG-Splatting uses fixed axes in SG and mixes SHs with SGs, resulting in a heavier and more complex representation with larger memory consumption."
> > > > >
> > > > > To avoid potential confusion, we have revised the contribution statement in the final manuscript. We have explicitly credited SG-Splatting for the initial exploration of SGs while clarifying that our specific contribution lies in achieving a fully standalone SG representation that enables efficient VRAM reduction without relying on hybrid models.

---

### Author Response · Authors · 2025-12-03
**Summary for the discussion period**

## Summary for the discussion period

We thank the reviewers for their constructive feedback. Reviewers FDsv and 1h2z strongly support our work. Additionally, Reviewer RnYG rated Soundness, Presentation, and Contribution as "3: good". Notably, Reviewer zwHD engaged in three rounds of active discussion, confirming that *"All of my questions have been resolved"* regarding technical details and finally stating *"I am satisfying all other questions but Q1"*. Below, we summarize the concerns and our responses.

---

### Main contribution

In this paper, we identify the 'low storage but high VRAM' issue in existing methods and shift the focus from the prevalent storage compression to rendering VRAM compression. Our key technical contributions are twofold:

1. **Unified pruning framework (primitive-wise \+ lobe-wise):** We propose a unified soft pruning framework that models both primitive-count and lobe-count as a memory-constrained optimization problem, which yields superior performance compared to existing staged or hard-pruning methods.

2. **Standalone SG Representation:** Unlike the hybrid SG-Splatting (Wang et al., 2024a), we fully replace SH with arbitrarily-oriented and prunable SGs, significantly lowering VRAM via reduced per-primitive parameters.


---

### Main Issues and the Discussion

The rebuttal discussions primarily revolved around these two technical contributions:

**1\. Claims regarding SG contribution (Contribution 1\) comparing SG-Splatting  (zwHD, RnYG, vVWJ)**

We clarified this issue from two key aspects:

* Our VRAM reduction stems not only from replacing SH with SG but crucially from pruning SGs, a direction unexplored by prior works like SG-Splatting.
* SG-Splatting employs a hybrid model (SH \+ fixed-axis SG), resulting in a heavier representation where the fixed-axis SGs lack expressivity and do not support lobe-wise pruning. In contrast, we achieve a complete replacement using arbitrarily-oriented SGs, which offer superior expressivity and are naturally compatible with lobe-wise pruning.

Through a deep discussion with Reviewer zwHD on this matter, we corrected misconceptions regarding SG-Splatting (e.g., storage vs. VRAM) and accepted the suggestion to refine the contribution statement. Accordingly, we revised the manuscript to explicitly credit its pioneering role while emphasizing our distinct "standalone, arbitrarily-oriented, and prunable" SG representation.

**2\. Clarifications on misunderstandings of the unified pruning framework (zwHD, vVWJ)**

We believe the initial concerns stemmed from overlooking our simultaneous primitive-wise and lobe-wise pruning, which led to the misconception that we merely introduced sharpness as an auxiliary metric. We clarified our contribution by:

1. Generalizing the problem from "minimizing primitive count" to "minimizing total VRAM cost", which is achieved by both reducing primitive numbers and reducing internal parameters (pruning SG lobe).
2. Detailing the VRAM cost formalization and the distinct roles of opacity and sharpness in our framework.

This clarification was well-received by Reviewer zwHD (Nov 25), who stated that *"the reformulation of GaussianSpa from the perspective of memory consumption is a smart idea"* and confirmed that *"these questions are now clear to me"*.

---

### Resolution of Other Concerns

We have addressed all specific requests from reviewers. Below, we highlight the major additional experiments, followed by a summary of other clarifications.

**Major Additional Experiments & Data:**

* **Comparison with additional SOTA methods (Ref. RnYG, vVWJ)**
  We added comparisons with LocoGS, LightGaussian, and MaskGaussian. MEGS² consistently achieves the lowest VRAM (\~40% lower than GaussianSpa)  compared to these baselines while maintaining comparable or better quality.

* **Requests for Detailed Metrics (Ref. zwHD, 1h2z, RnYG, vVWJ):**
  We reported all requested metrics, addressing concerns about storage size (Table 10), primitive counts (Table 13), training time (Table 12), and average lobe usage (Figure 5).






**Other Clarifications & Discussions:**

We have comprehensively addressed all remaining inquiries, spanning methodology details (e.g., post-processing, hyperparameters), systematic analysis (e.g., VRAM composition, sensitivity tests), visualizations (e.g., ablations, extreme scenarios), and discussions on future work.

Notably, Reviewer zwHD engaged in three rounds of active discussion, explicitly validating our responses by describing our reformulation as a "smart idea" and confirming "these questions are now clear." In response to the final note, "I am satisfying all other questions but Q1," we addressed this last point in our Nov 28 response with a detailed clarification and a corresponding manuscript revision. We hope these efforts demonstrate the comprehensiveness of our revisions.

---

### Meta-Review · Area_Chair_htBb · 2026-01-07

**Summary:**

The review ratings for this paper are quite divergent. The major concerns are mainly around the contributions against SG-Splatting and GS-Spa as well as technical details and experiment results. The authors did a good job in the rebuttal. Although acknowledging inaccurate claims, they managed to explain the main difference with SG-Splatting in terms of the color representation, where SG-Splating uses a mix of SH and SG while this paper introduces arbitrarily-oriented SG alone that facilitates subsequent lobe pruning. Compared to GS-Spa, this work prunes not only GSs but also SG lobes jointly. Overall, the AC feels the contributions are significantly and the saving the VRAM is impressive.

**Reviewer Concerns:**

See above.

**Reviewer Scores:**

I think R1 and R4 are likely to upgrade their review ratings since the authors' rebuttal well addressed their concerns.

---

### Decision · Program_Chairs · 2026-01-26

Accept (Poster)